# Multi-Synaptic Cooperation: A Bio-Inspired Framework for Robust and Scalable Continual Learning

**Penghui Li**[1,2]* **Zhuang Ma**[1,3]* **Yunliang Zang**[4]† **Qiang Yu**[1,3]†

[1] School of Artificial Intelligence, CCA Lab, Tianjin University, Tianjin, China
[2] School of Future Technology, Tianjin University, Tianjin, China
[3] College of Intelligence and Computing, Tianjin University, Tianjin, China
[4] Academy of Medical Engineering and Translational Medicine, Tianjin University, Tianjin, China
{lipenghui, mazhuang, yunliangzang, yuqiang}@tju.edu.cn

## Abstract

Continual learning aims to acquire new knowledge incrementally while retaining prior information, with catastrophic forgetting (CF) being a central challenge. Existing methods can mitigate CF to some extent but are constrained by limited capacity, which often requires dynamic expansion for long task sequences and makes performance sensitive to task order. Inspired by the richness and plasticity of synaptic connections in biological nervous systems, we propose the Multi-Synaptic Cooperation Network (MSCN), a generalized framework that models cooperative interactions among multiple synapses through multi-synaptic connections modulated by local synaptic activity. This design enhances model representational capacity and enables task-adaptive plasticity by means of multi-synaptic cooperation, providing a new avenue for expanding model capacity while improving robustness to task order. During learning, our MSCN dynamically activates task-relevant synapses while suppressing irrelevant ones, enabling targeted retrieval and minimizing interference. Extensive experiments across four benchmark datasets, involving both spiking and non-spiking neural networks, demonstrate that our method consistently outperforms state-of-the-art continual learning methods with significantly improved robustness to task-order variation. Furthermore, our analysis reveals an optimal trade-off between synaptic richness and learning efficiency, where excessive connectivity can impair circuit performance. These findings highlight the importance of the multi-synaptic cooperation mechanism for achieving efficient continual learning and provide new insights into biologically inspired, robust, and scalable continual learning.

## 1 Introduction

Continual learning aims to develop models capable of acquiring and retaining knowledge from a sequence of tasks or data distributions, thereby mimicking the human ability to learn progressively over time. This approach is also known as lifelong learning or incremental learning Thrun (1995) and holds promise for building adaptive and efficient systems in dynamic environments. A core challenge in continual learning is catastrophic forgetting McCloskey & Cohen (1989)—a phenomenon where the model's performance on previously acquired tasks degrades significantly when updated with new data De Lange et al. (2021); Parisi et al. (2019); Masana et al. (2023).

Recently, various approaches have been proposed to mitigate catastrophic forgetting Bonicelli et al. (2022); Tong et al. (2022); Qiao et al. (2024); Li et al. (2024b). These approaches can be broadly categorized into three primary types Masana et al. (2023): *Rehearsal-based* methods Lopez-Paz & Ranzato (2017); Bang et al. (2021); Van de Ven et al. (2020); Hayes et al. (2020), *Regularization-based* methods Kirkpatrick et al. (2017); Wołczyk et al. (2022); Schwarz et al. (2018); Li et al.

---

*Equal contribution. Both authors were graduate students under the supervision of Prof. Q. Yu.
†Corresponding author.

(a). Richness of Synaptic Connections

(b). Multiple Synapses Connect Neurons

(c). Modulation Based on Eligibility Trace

(d). Selective Activation and Inhibition of Synapses

Figure 1: Overview of the MSCN framework. (a) The phenomenon of multi-synaptic connections in biological neurons. (b) Modeling of neurons with multiple synaptic connections in both SNNs and ANNs. (c) Local synaptic activity plasticity based on eligibility traces. Depending on the eligibility trace, synapses undergo potentiation (P), depression (D), or remain unchanged (O), reflecting the modulation's strength and direction. (d) The process of selectively activating task-relevant synapses and inhibiting irrelevant ones.

(2024c), and *Architecture-based* methods Yoon et al. (2017); Zhou et al. (2021); Li et al. (2019); Wang et al. (2022a); Serra et al. (2018); Hung et al. (2019); Kang et al. (2022a). Among these three types, *Architecture-based* methods are particularly notable for the ability to dynamically adjust the network structure to accommodate new tasks. They primarily rely on two strategies: network expansion and pruning. Expansion strategies Hung et al. (2019); Li et al. (2019); Yoon et al. (2017) start with a small model and dynamically expand the network to mitigate forgetting. In contrast, pruning strategies Mallya et al. (2018); Wang et al. (2022b); Kang et al. (2022a) assign a sub-network to each old task. These sub-networks are pruned from a pre-allocated dense model. Further training is restricted to the unpruned parameters only. While these strategies offer several advantages, including performance and the potential for forget-free learning in some cases, they still face notable challenges: (i) Dynamic expansion requires continuous network growth to accommodate new tasks, which makes it hardware-unfriendly; (ii) Both expansion and pruning methods are often sensitive to the task order, leading to significant performance variation depending on the sequence of tasks.

Remarkably, the brain achieves continual learning without suffering from dynamic structural growth Rasch & Born (2007); Joseph & Gu (2021), highlighting the potential of biologically inspired mechanisms as significant alternatives to artificial network expansion. Among these, multi-synaptic connectivity is believed to be a key factor in supporting continual learning Shi et al. (2025); Wu & Mel (2009); Ko et al. (2011), as its diverse synaptic architecture enhances the information representation capacity of neural circuits. Meanwhile, *Three-factor* learning rules Frémaux & Gerstner (2016); Gerstner et al. (2018) have been widely studied as biologically plausible models of synaptic plasticity, in which synaptic changes are not only modulated by global neuromodulatory signals but also depend on the local synaptic activity. Inspired by this, we propose MSCN, a novel framework that enhances representational flexibility and robustness by employing the multi-synaptic cooperation mechanism, rather than increasing network depth, width, or dynamically expanding the architecture, as illustrated in Fig. 1. Our framework consists of two components: the multi-synapse connectivity structure that augments the model's representational capacity within a fixed network architecture, and the synaptic plasticity modulation mechanism based on local synaptic activity. Such local synaptic activity is integrated via eligibility traces, which serve as modulatory signals to synaptic weight

updates. During learning, task-relevant synapses are dynamically selected, while irrelevant ones are suppressed, thereby effectively minimizing interference across tasks.

Our contributions are as follows:

- We propose MSCN, the first continual learning framework that explicitly leverages the multi-synaptic cooperation mechanism, providing a biologically inspired and capacity-efficient solution. By maximally harnessing synaptic resources, MSCN unlocks the potential of fixed-capacity networks and substantially boosts the scalability of continual learning systems.

- We design a modulatory mechanism based on local synaptic activities that modulates synaptic plasticity through eligibility traces, enabling precise, activity-dependent modulation at the synaptic level. This modulation significantly strengthens the robustness of continual learning models to task order variations, ensuring stable performance even under highly dynamic and unpredictable training sequences.

- Extensive experiments on four benchmark datasets across both spiking and non-spiking architectures demonstrate that MSCN consistently outperforms state-of-the-art continual learning methods in terms of accuracy, forgetting mitigation, and robustness to task order, while also exhibiting competitive computational efficiency.

## 2 RELATED WORK

**Continual Learning** methods are roughly divided into three categories: *Rehearsal-based* methods store past experiences in memory to mitigate forgetting. Some works Rebuffi et al. (2017); Tiwari et al. (2022); Zhou et al. (2022b); Jeeveswaran et al. (2023) design sampling strategies to allocate a limited memory budget, while others Lin et al. (2022); Rolnick et al. (2019); Sun et al. (2023) build special subspace of old tasks as the memory. *Regularization-based* methods aim to consolidate previous knowledge by adding extra regularization terms to the loss function. Some works Li & Hoiem (2017); Kirkpatrick et al. (2017); Cha et al. (2020) constrain important weights in the parameter space Akyürek et al. (2021); Rudner et al. (2022); Kim et al. (2023), feature representations Gao et al. (2022); Jeeveswaran et al. (2023), or output logits Li & Hoiem (2017); Oh et al. (2022) to remain close to those of the old model. *Architecture-based* methods dedicate different incremental model structures towards each task to minimize forgetting Zhou et al. (2022a); Lu et al. (2024); Kang et al. (2022a). Some works Serra et al. (2018); Yoon et al. (2019); Hu et al. (2023) adopt modular architectures by dynamically expanding additional components Yan et al. (2021); Zhu et al. (2022), or freeze subsets of parameters Abati et al. (2020); Liu et al. (2021) to overcome forgetting. In this work, to better investigate the capacity efficiency and robustness of our method, we implement our multi-synaptic cooperation mechanism based on the architecture-based methods without dynamically expanding the network.

**Neural Network Dynamics** in continual learning describe how internal representations and connectivity patterns evolve as new tasks are learned sequentially Márton et al. (2022). These dynamics manifest across multiple levels, including synaptic updates, activation trajectories, and parameter plasticity under task transitions Vyas et al. (2020). Recent works have begun to explicitly model neural dynamics in continual learning by introducing mechanisms such as synaptic trajectories, context-dependent modulation, and task-driven weight routing Li & Wang (2017); Li et al. (2024a); Xu et al. (2024). These studies highlight the importance of capturing temporal evolution in network parameters to support adaptive behavior over extended task sequences. Additional research explores dynamic mechanisms such as gating, masking, and sparsity-inducing priors to modulate parameter updates and isolate task-specific pathways Abati et al. (2020); Wang et al. (2022c); Yan et al. (2022). Recently, multi-synaptic (redundant) connections between neuron pairs have been shown to enhance computational capacity Zenke & Laborieux (2024); Hofmann et al. (2025). Nevertheless, the cooperative interactions among multiple synapses and their implications for continual learning remain largely underexplored. In contrast, our method introduces multi-synaptic dynamics within each connection and the modulation based on local synapse activity, enabling representational diversity and adaptive modulation of synaptic plasticity. This novel design does not require increasing network depth/width or dynamically expanding the network; instead, it increases capacity and enhances robustness through a multi-synaptic cooperation mechanism.

## 3 METHOD

### 3.1 MODELING MULTI–SYNAPTIC SPIKING NEURON

Biological systems achieve continual learning without relying on architectural growth Song et al. (2024); Shi et al. (2025). A key neurobiological observation is the presence of multiple synaptic contacts between the same axon–dendrite pair, providing redundancy and adaptability Trachtenberg et al. (2002); Yang et al. (2014). Since our design operates at the synaptic level, the proposed method can be applied to both ANN and SNN Maass (1997); Gütig & Sompolinsky (2006); Yu et al. (2025) architectures. To better capture biological principles, we first model the multi-synaptic cooperation mechanism in SNNs, which are more consistent with biological processes and offer event-driven, temporally sparse, and energy-efficient computation Gütig & Sompolinsky (2006). As a concrete instantiation, we adopt the leaky integrate-and-fire (LIF) neuron Lapicque (1907), a widely used model balancing biological plausibility and computational simplicity Shiu et al. (2024); Brand & Petruccione (2024). The membrane potential $V(t)$ evolves over continuous time as follows:

$$\tau_{\mathrm{m}} \frac{dV(t)}{dt} = -\big(V(t) - V_{\mathrm{rest}}\big) + I(t) \tag{1}$$

where $\tau_{\mathrm{m}}$ is the membrane time constant, $V_{\mathrm{rest}}$ is the resting potential, and $I(t)$ denotes the total synaptic input current. This formulation captures leakage and current integration but assumes a single synapse per connection, limiting diversity. To address this, we generalize the neuron model by introducing $P \geq 1$ parallel synapses for each synaptic connection. Consider a neuron with $N$ presynaptic neurons, each forming $P$ distinct synaptic pathways to it. Therefore, in continuous time, the membrane potential of the postsynaptic neuron is given by:

$$V(t) = \sum_{i=1}^{N} \sum_{p=1}^{P} w_{ip} \mathrm{PSP}_{ip}(t) - \vartheta \sum_{j} e^{-\frac{t - t_s^j}{\tau_{\mathrm{m}}}} \tag{2}$$

where $w_{ip}$ denotes the synaptic weight of the $p$-th parallel synapse associated with the $i$-th presynaptic neuron, $\mathrm{PSP}_{ip}$ represents the postsynaptic potential of this synapse and $\vartheta$ denotes the firing threshold. To preserve synaptic heterogeneity Deng et al. (2025) and enable independent optimization, we introduce distinct decay time constants across parallel synapses. Accordingly, for the $p$-th synapse of presynaptic neuron $i$, the spike arrival times are denoted by $t_{ip}^f$, and $\mathrm{PSP}_{ip}$ is defined as:

$$\mathrm{PSP}_{ip}(t) = \sum_{f} K_{ip}\big(t - t_{ip}^f\big) \tag{3}$$

$$K_{ip}(t) = e^{-\frac{t}{\tau_{s_{ip}}}} \tag{4}$$

where $K_{ip}(t)$ denotes the kernel function of the $p$-th parallel synapse, and $\tau_{s_{ip}}$ represents its decay time constant, which is initialized to different values (non-trainable). This design allows multiple temporal and weighted channels to influence the synaptic plasticity, thereby enhancing the diversity of spatiotemporal representations. On this basis, modeling in the ANN architecture can be more readily formulated. Given space limitations, the corresponding implementation is presented in Appendix A.1.

### 3.2 PLASTICITY MODULATION BASED ON ELIGIBILITY TRACES

Building on multi-synaptic connections, we introduce a modulation mechanism of synaptic plasticity based on local synaptic activity. Specifically, we propose the eligibility trace as the basis for modulating local synaptic plasticity. We begin by formulating the modulation signal in continuous time as a cumulative sum of synaptic spike events. For a connection between two neurons, $P$ parallel synapses share a common eligibility trace, which is defined as:

$$\frac{d\tilde{e}}{dt} = -\frac{\tilde{e}}{\tau} + \sum_{f} \delta\big(t - t^f\big) \tag{5}$$

where $\tau$ is the decay time constant, $\delta(\cdot)$ denotes the Dirac delta function. For practical implementation, we adopt a discrete-time formulation, and the dynamics of $\tilde{e}$ are updated as follows:

$$\tilde{e}[t+1] = \tilde{e}[t] - \frac{\tilde{e}[t]}{\tau} + S[t+1] \tag{6}$$

where $S[t+1] \in \{0,1\}$ indicates whether a spike occurred at time step $t+1$. This design captures recent spiking accumulation while allowing the eligibility trace to decay in the absence of input, enabling the eligibility trace to track local synaptic activity over time.

The effect of the modulation factor on synaptic plasticity is governed by a nonlinear function $f_{\mathrm{mod}}(\tilde{e})$, which determines how strongly and in what direction the synapse should change in response to local synaptic activity signals. Following Zhang et al. (2023), we adopt a piecewise quadratic form for $f_{\mathrm{mod}}$, which adjusts the strength and direction (potentiation or depression) of synaptic plasticity depending on the eligibility trace. In practice, $\tilde{e}$ is normalized over all model synapses to $[-1, 1]$ to reflect the relative strength of local synaptic activity. The modulation function is defined as:

$$f_{\mathrm{mod}}(\tilde{e}) = \begin{cases} 1 + (\theta_{\max} - 1)\left(\dfrac{|\tilde{e}|}{e_1}\right)^2, & 0 \leq |\tilde{e}| \leq e_1, \\[2mm] \theta_{\max}\left[1 - \left(\dfrac{|\tilde{e}| - e_1}{e_{\mathrm{inv}} - e_1}\right)^2\right], & e_1 \leq |\tilde{e}| \leq e_{\mathrm{inv}}, \\[2mm] -f_{\mathrm{mod}}(2e_{\mathrm{inv}} - |\tilde{e}|), & e_{\mathrm{inv}} \leq |\tilde{e}| \leq 2e_{\mathrm{inv}}, \\[2mm] 0, & otherwise \end{cases} \tag{7}$$

where $e_1 = 0.5\, e_{\mathrm{inv}}$ sets the low–moderate boundary, $\theta_{\max}$ controls the maximum modulation strength, and $e_{\mathrm{inv}}$ marks the zero point of $f_{\mathrm{mod}}$. (Fig. 1c). This function modulates plasticity based on local synaptic activity: potentiation occurs when $|f_{\mathrm{mod}}| \geq 1$, depression arises when $0 < |f_{\mathrm{mod}}| < 1$, and complete depression is observed when $f_{\mathrm{mod}} = 0$. The final modulated synaptic weight change is computed as:

$$\Delta w = -\eta \cdot \left| f_{\mathrm{mod}}(\tilde{e}) \right| \cdot \frac{\partial \mathcal{L}}{\partial w} \tag{8}$$

where $\eta$ is the learning rate and $\partial \mathcal{L}/\partial w$ is the gradient. The modulation function $f_{\mathrm{mod}}$ adjusts the size and direction of the synaptic weight update. By dynamically adjusting synaptic updates in response to recent local activity, this design mimics the biological mechanism of robust learning through activity-dependent modulation Wu et al. (2021); Wu & Maass (2025), thereby enabling the network to adapt to changing task demands and maintain stable performance.

### 3.3 GENERALIZING TO CLASSIC ARCHITECTURE-BASED METHODS

In this section, we integrate our method into the *Architecture-based* setting, adopting the same setup as in Kang et al. (2022b); Serra et al. (2018); Wortsman et al. (2020). Consider a standard supervised continual learning setting with $T$ tasks presented sequentially. For each task $j$, the model receives a dataset $\mathcal{D}_j = \{(\mathbf{x}_{i,j}, y_{i,j})\}_{i=1}^{n_j}$ consisting of $n_j$ labeled samples. A fixed-topology deep neural network $\mathcal{F}(\cdot; \boldsymbol{\theta})$, parameterized by model parameters $\boldsymbol{\theta}$, is employed. The objective at each step is to optimize the model for the current task $j$:

$$\boldsymbol{\theta}^* = \underset{\boldsymbol{\theta}}{\mathrm{minimize}} \; \frac{1}{n_j} \sum_{i=1}^{n_j} \mathcal{L}\left(\mathcal{F}(\mathbf{x}_{i,j}; \boldsymbol{\theta}), y_{i,j}\right) \tag{9}$$

Following Gao et al. (2023); Wortsman et al. (2020), task identities are assumed to be available during both training and inference, under a multi-head setting in which each task is assigned a distinct output head. For each task $j$, a binary mask $\mathbf{m}_j^*$ is learned to activate the relevant synapses. The training objective is formulated as:

$$\boldsymbol{\theta}^*, \mathbf{m}_j^* = \underset{\boldsymbol{\theta}, \mathbf{m}_j}{\text{minimize}} \frac{1}{n_j} \sum_{i=1}^{n_j} \Big[ \mathcal{L}\big(\mathcal{F}(\mathbf{x}_{i,j}; \boldsymbol{\theta} \odot \mathbf{m}_j), y_{i,j}\big) - \mathcal{L}\big(\mathcal{F}(\mathbf{x}_{i,j}; \boldsymbol{\theta}), y_{i,j}\big) \Big] \qquad (10)$$

where $\odot$ denotes element-wise multiplication. A shared learnable relevance score $\mathbf{r}$ is maintained across tasks, with each entry corresponding to a synapse Kang et al. (2022b). Trained jointly with the network parameters, $\mathbf{r}$ enables the model to identify task-relevant connections. For task $j$, the subnetwork $\hat{\boldsymbol{\theta}}_j$ is formed by selecting the top $c\%$ of weights ranked by relevance, where $c$ is the layerwise capacity ratio Wortsman et al. (2020). The selected weights are indicated by the binary mask $\boldsymbol{m}_j$, in which a value of 1 signifies that the corresponding weight is active during the forward pass, and 0 indicates it is deactivated. To preserve past knowledge, an accumulated mask $\mathbf{M}_{j-1} = \bigvee_{i=1}^{j-1} \mathbf{m}_i$ (with $\bigvee$ as logical OR) is applied when learning task $j$. The parameters $\boldsymbol{\theta}$ are updated as:

$$\boldsymbol{\theta} \leftarrow \boldsymbol{\theta} - \boldsymbol{\Delta\theta} \odot (\mathbf{1} - \mathbf{M}_{j-1}) \qquad (11)$$

where $\boldsymbol{\Delta\theta}$ denotes the gradient step, and the term $(\mathbf{1} - \mathbf{M}_{j-1})$ ensures that only unallocated synapses remain trainable, thereby preserving the stability of parameters from previously learned tasks.

## 4 EXPERIMENTS

### 4.1 EXPERIMENTAL SETUP

We conduct comprehensive experiments under diverse training configurations, input domains, datasets, and network architectures, and evaluate continual learning performance using two widely adopted metrics Kang et al. (2022a); Konishi et al. (2023): *Average Accuracy (ACC)* and *Backward Transfer (BWT)*. ACC measures the model's average accuracy over all tasks after learning task $T$, reflecting its overall generalization ability. BWT measures the impact of new tasks on prior ones, with higher values better and 0 indicating no forgetting. Unless otherwise stated, we set the synapse count to $P = 3$ in all main experiments, ablation studies, and computational cost analyses. Implementation and hardware details are provided in Appendix A.2.

Table 1: Performance comparison on four datasets, evaluating performance under both SNN and ANN frameworks. We report the results across 5 independent runs with different random seeds under the same experimental setup. Table 5 in Appendix A.2 shows the standard deviations.

| Network | Method | PMNIST | | 10-split CIFAR-100 | | TinyImageNet | | 5-Datasets | |
|---|---|---|---|---|---|---|---|---|---|
| | | ACC (%) ↑ | BWT (%) ↑ | ACC (%) ↑ | BWT (%) ↑ | ACC (%) ↑ | BWT (%) ↑ | ACC (%) ↑ | BWT (%) ↑ |
| SNN | *MTL* | 96.52 | / | 79.83 | / | 79.24 | / | 89.93 | / |
| | EWC[PNAS] Kirkpatrick et al. (2017) | 91.45 | -3.20 | 73.75 | -4.89 | 60.29 | -25.47 | 57.06 | -44.55 |
| | HAT[ICML] Serra et al. (2018) | 93.25 | -2.07 | 73.67 | -0.13 | 62.18 | -8.51 | 72.72 | -22.90 |
| | GPM[ICLR] Saha et al. (2021) | 94.80 | -1.62 | 77.48 | -1.37 | 70.07 | -2.92 | 79.70 | -15.52 |
| | HLOP[ICLR] Xiao et al. (2024) | 95.15 | -1.30 | 78.58 | -0.26 | 71.40 | -0.52 | 88.65 | -3.71 |
| | **MSCN** | **96.34** | **0.0** | **79.54** | **0.0** | **73.22** | **0.0** | **88.84** | **0.0** |
| ANN | EWC[PNAS] Kirkpatrick et al. (2017) | 92.01 | -0.03 | 72.77 | -3.59 | 64.51 | -0.04 | 88.64 | -0.04 |
| | GPM[ICLR] Saha et al. (2021) | 94.96 | -0.02 | 73.18 | -1.17 | 67.39 | 1.45 | 91.22 | -0.01 |
| | PackNet[CVPR] Mallya et al. (2018) | 96.37 | 0.0 | 72.39 | 0.0 | 55.46 | 0.0 | 92.81 | 0.0 |
| | SupSup[NeurIPS] Wortsman et al. (2020) | 96.31 | 0.0 | 75.47 | 0.0 | 59.60 | 0.0 | 93.28 | 0.0 |
| | WSN[ICML] Kang et al. (2022a) | 96.41 | 0.0 | 76.38 | 0.0 | 71.96 | 0.0 | 93.41 | 0.0 |
| | TAMiL[ICLR] Bhat et al. (2023) | 96.87 | -3.15 | 76.73 | -3.47 | 72.55 | -3.02 | 93.47 | -4.72 |
| | SPG[ICML] Konishi et al. (2023) | 96.35 | – | 74.82 | – | 73.26 | – | 93.32 | – |
| | DFGP[ICCV] Yang et al. (2023) | 94.64 | -0.01 | 74.59 | 0.0 | – | – | 92.09 | -0.01 |
| | Bayesian[ICML] Thapa & Li (2024) | 96.74 | – | 75.57 | – | 73.93 | – | 93.36 | – |
| | **MSCN** | **97.53** | **0.0** | **77.37** | **0.0** | **75.03** | **0.0** | **93.69** | **0.0** |

### 4.2 COMPARISON TO THE STATE-OF-THE-ART METHODS

We first conduct a comprehensive evaluation of MSCN in a multi-head task-incremental learning scenario, using four widely used benchmark datasets and employing both SNNs and ANNs. As summarized in Table 1, where the dataset complexity roughly increases from left to right, the results consistently demonstrate the strength and reliability of MSCN across diverse datasets and

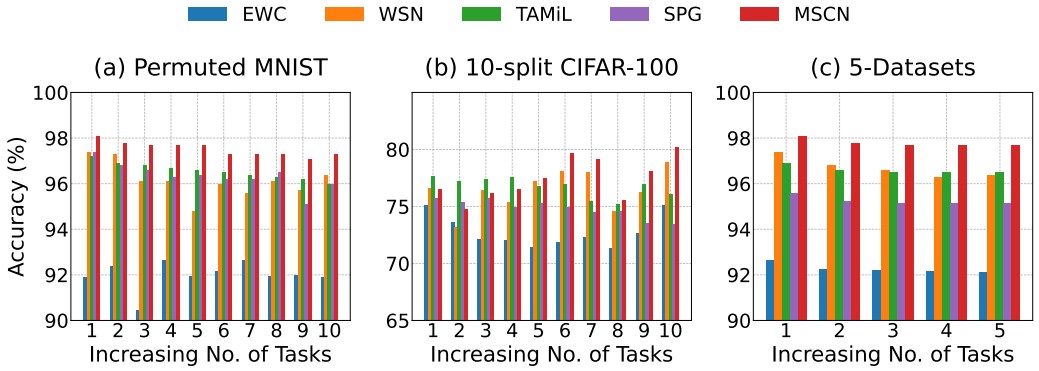

Figure 2: Per-task accuracy across the entire incremental learning.

architectures. Notably, on TinyImageNet with ANN, MSCN achieves an accuracy that exceeds the second-best method, Bayesian Thapa & Li (2024), by 1.10%. Although several approaches, such as WSN Kang et al. (2022a), SupSup Wortsman et al. (2020), and PackNet Mallya et al. (2018), achieve zero backward transfer, they do not match the overall accuracy of MSCN. These findings indicate that multi-synaptic connectivity is an effective and scalable design for continual learning. Moreover, Fig. 2 shows that MSCN attains superior per-task performance on most tasks, further validating its representational strength. Since most existing works on continual learning are based on ANN architectures, we investigate our method extensively within the same framework in the following sections to ensure fairness while highlighting its strengths.

## 4.3 ROBUSTNESS TO TASK ORDER PERMUTATIONS

We evaluate order robustness by training on multiple CIFAR-100 Split permutations and measuring per-task accuracy variation over three task orders. As illustrated in Fig. 3d, we observe that EWC Kirkpatrick et al. (2017) and GPM Saha et al. (2021) display large fluctuations across task sequences, highlighting their strong sensitivity to order. WSN Kang et al. (2022a) performs competitively with EWC but shows a tendency to overfit to particular task orders (Fig. 3a–b). In contrast, MSCN achieves stable accuracy across all tasks and permutations, with only minimal variation (Fig. 3c). This consistency indicates that MSCN can flexibly adapt to new tasks while mitigating interference and preserving prior knowledge, highlighting the important role of local activity-dependent synaptic modulation as a foundational mechanism for building scalable continual learning systems. Additional experiments on five task orders are reported in Fig. 10 of the Appendix A.3.4.

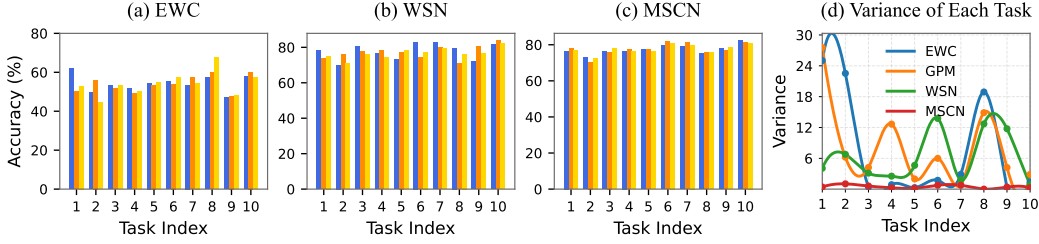

Figure 3: Task order robustness comparison on CIFAR-100 Split. Bar plots in (a), (b), and (c) show per-task accuracy under three different task sequences. Specifically, (a) corresponds to the EWC method, (b) to the WSN method, and (c) to our MSCN method. (d) shows the standard deviation of the accuracy for each task across different task sequences.

## 4.4 ABLATION STUDY

To better understand the individual contributions of the key components in MSCN, we perform an ablation study by selectively removing the multi-synaptic connectivity structure and the neuromodulatory mechanism. The results across four benchmark datasets are summarized in Table 2. The full MSCN model, with both components enabled, consistently achieves the highest accuracy across all datasets. Disabling the neuromodulatory mechanism results in a noticeable performance drop, particularly on TinyImageNet and PMNIST. Conversely, removing the multi-synaptic structure also results in performance degradation, particularly on CIFAR-100 and 5-Datasets. We observe that when both components are ablated, performance drops further across all datasets, confirming that the two mechanisms act cooperatively and that their interaction is essential for MSCN's ability to achieve robust and scalable continual learning.

Table 2: An ablation study of MSCN on ACC. ✓indicates that the component is included, while ✗indicates that it is excluded.

| Multi-synapse | Modulation | PMNIST | CIFAR-100 | TinyImageNet | 5-Datasets |
|:---:|:---:|:---:|:---:|:---:|:---:|
| ✓ | ✓ | 97.53 (± 0.19) | 77.37 (± 0.23) | 75.03 (± 0.27) | 93.69 (± 0.21) |
| ✗ | ✓ | 96.79 (± 0.20) | 77.03 (± 0.22) | 73.81 (± 0.26) | 93.47 (± 0.24) |
| ✓ | ✗ | 96.53 (± 0.21) | 76.81 (± 0.25) | 73.78 (± 0.29) | 93.51 (± 0.22) |
| ✗ | ✗ | 96.34 (± 0.22) | 76.34 (± 0.24) | 72.59 (± 0.31) | 93.32 (± 0.25) |

## 4.5 EFFECT OF SYNAPSE COUNT

To probe the role of multi-synaptic connectivity, we vary synapse and neuron counts and measure performance across tasks. On CIFAR-100 Split (Fig. 4), increasing synapses per connection consistently raises per-task accuracy as the number of tasks grows. Jointly scaling synapses and neurons (Fig. 5) reveals a saturation regime: accuracy improves with capacity but plateaus once model capacity exceeds task complexity. Fig. 6 summarizes four benchmarks; rows (top→bottom) are PMNIST, CIFAR-100, TinyImageNet, and 5-Datasets, reflecting increasing complexity. We observe that, although the optimal number of synapses varies with task difficulty, it generally stabilizes once a certain threshold is exceeded. Interestingly, the observed performance trend happens to mirror how synaptic counts are distributed in the brain—typically confined to a limited but effective rangeToni et al. (1999); Watson et al. (2025). Additional experimental results are provided in Appendix A.3.3.

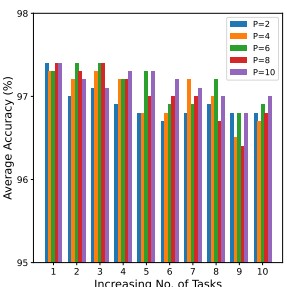 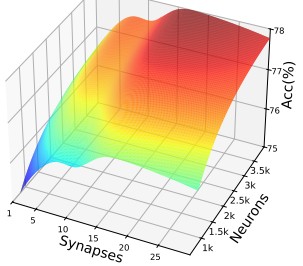 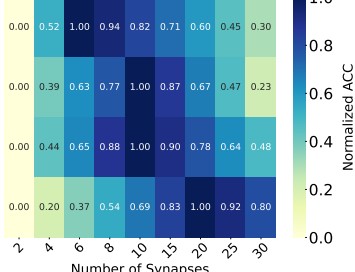

Figure 4: Per-task accuracy under different synapse counts on CIFAR-100 Split.

Figure 5: Accuracy under different synapse and neuron counts on CIFAR-100 Split.

Figure 6: Accuracy changes across four datasets under varying numbers of synapses

## 4.6 CAPACITY ANALYSIS

To evaluate the capacity efficiency of MSCN, we adopt the commonly used metric CAP Kang et al. (2022a); Wortsman et al. (2020) (defined in Appendix A.2.2) and compare it with baseline approaches, where lower CAP values indicate better capacity utilization. Fig. 7 shows the relationship between accuracy and total capacity usage across four benchmark datasets. The results demonstrate that MSCN

consistently achieves the highest accuracy with significantly lower capacity overhead. On Permuted MNIST and TinyImageNet (Fig. 7a,c), MSCN outperforms all baselines while requiring substantially fewer resources. Similar trends are observed on CIFAR-100 and the 5-Datasets benchmark (Fig. 7b,d), where methods such as PackNet Mallya et al. (2018) and SupSup Wortsman et al. (2020) consume much more capacity but yield lower or comparable performance. These results indicate that the multi-synaptic cooperation mechanism enables more effective knowledge storage and reuse across tasks.

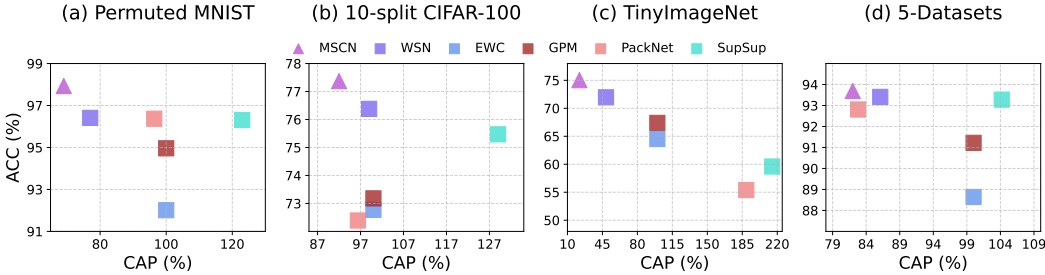

Figure 7: Accuracy over CAP (%) across four benchmarks.

## 4.7 COMPUTATIONAL EFFICIENCY

Since multi-synaptic connections inevitably increase the parameter count, we conducted experiments under the same parameter budget as the baselines to fairly evaluate computational efficiency. We further evaluated robustness under five task orders using AOPD (where lower values indicate stronger robustness; see Appendix A.2.2 for details). As shown in Table 3, although reducing the parameter count leads to a drop in MSCN's accuracy, our method consistently achieves the best efficiency and robustness while maintaining competitive accuracy. These results demonstrate that the proposed multi-synaptic cooperation mechanism achieves high computational efficiency and establishes the foundation for robust and scalable continual learning.

Table 3: Computational efficiency under the same parameter budget (Training time in hours).

| Method | PMNIST | | | 10-split CIFAR-100 | | |
|---|---|---|---|---|---|---|
| | Training Time ↓ | ACC (%) ↑ | AOPD (%) ↓ | Training Time ↓ | ACC (%) ↑ | AOPD (%) ↓ |
| PackNet | 0.59 (± 0.15) | 96.43 (± 0.18) | 2.23 | 1.13 (± 0.10) | 72.45 (± 0.20) | 5.36 |
| SupSup | 0.53 (± 0.12) | 96.36 (± 0.22) | 1.27 | 0.87 (± 0.08) | 75.54 (± 0.17) | 3.81 |
| WSN | 0.38 (± 0.05) | 96.49 (± 0.13) | 0.29 | 0.78 (± 0.06) | 76.47 (± 0.34) | 2.59 |
| **MSCN** | **0.33** (± **0.11**) | **96.89** (± **0.19**) | **0.23** | **0.65** (± **0.04**) | **76.40** (± **0.14**) | **2.41** |

| Method | TinyImageNet | | | 5-Datasets | | |
|---|---|---|---|---|---|---|
| | Training Time ↓ | ACC (%) ↑ | AOPD (%) ↓ | Training Time ↓ | ACC (%) ↑ | AOPD (%) ↓ |
| PackNet | 1.45 (± 0.12) | 55.51 (± 0.25) | 6.51 | 3.45 (± 0.08) | 92.89 (± 0.12) | 4.37 |
| SupSup | 0.97 (± 0.07) | 59.65 (± 0.24) | 6.94 | 3.26 (± 0.10) | 93.31 (± 0.16) | 2.83 |
| WSN | 0.92 (± 0.04) | 72.03 (± 0.41) | 4.98 | 3.05 (± 0.08) | 93.50 (± 0.13) | 1.37 |
| **MSCN** | **0.75** (± **0.03**) | **74.04** (± **0.21**) | **4.73** | **2.82** (± **0.09**) | **93.33** (± **0.09**) | **1.26** |

## 5 CONCLUSION

In this paper, we propose MSCN, the first continual learning framework that explicitly models multi-synaptic cooperation. By equipping each connection with multiple plastic synapses and employing local synaptic activity–based modulation, MSCN achieves effective knowledge retention and adaptability within a fixed architecture. Extensive evaluations across diverse datasets and architectures demonstrate that our approach consistently outperforms state-of-the-art baselines in terms of computational efficiency and task order robustness. These findings highlight that the synergistic interplay

between multi-synaptic connectivity and localized plasticity modulation substantially enhances the network's representational capacity, providing new insights for robust and scalable continual learning. In addition, the similarity between our model's synapse counts and those found in biological systems further supports the plausibility of our design. Future work will explore allocating different numbers of synapses across neuron connections to further optimize synaptic resource utilization.

ACKNOWLEDGMENTS

This work was in part supported by the Brain Science and Brain-like Intelligence Technology - National Science and Technology Major Project (Grant No. 2025ZD0215600), the National Natural Science Foundation of China (Grant Nos. 92370103 and 62176179), and in part by the Xiaomi Foundation. We thank the reviewers for their insightful and supportive comments.

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

## A    APPENDIX

### A.1    MULTI-SYNAPTIC COOPERATION NETWORK IN ANNS

#### A.1.1    MODELING MULTI-SYNAPTIC NEURON

In biological neurons, a single axon can establish multiple synaptic contacts with the same dendritic branch, enabling the signal to influence the target neuron through parallel pathways Turegano-Lopez et al. (2024); Callan et al. (2021). To simulate this structure in artificial neural networks, we represent each inter-neuronal connection not by a single weight but by a vector of parallel synaptic weights. Each element in this vector is trained independently, and their combined effect determines the connection strength. This design remains compatible with existing network architectures, supports standard gradient-based learning, and naturally extends plasticity rules developed for spiking neural networks.

To model multi-synaptic connections in ANNs, we first associate each connection $(i, j)$ with a column vector of synaptic weights $\mathbf{w}_{ij}$:

$$\mathbf{w}_{ij} = \left[ w_{ij}^1, \ldots, w_{ij}^P \right]^\top \tag{12}$$

where $P$ is a hyperparameter representing the number of synaptic connections between each pair of neurons. To preserve synaptic diversity and enable distinct optimization of parallel synapses, we assign different activation functions to different parallel synapses. The input to postsynaptic neuron $j$ from presynaptic neuron $i$ is then defined as:

$$g_{ij}(x_i) = \sum_{p=1}^{P} \sigma_p\big(w_{ij}^p x_i\big) \tag{13}$$

where $\sigma_p$ denotes the activation function specific to the $p$-th synapse. When an input vector $\mathbf{x}$ is given to a fully connected layer, the input to neuron $j$ is computed as

$$z_j = \sum_{i=1}^{N} g_{ij}(x_i) = \sum_{i=1}^{N} \sum_{p=1}^{P} \sigma_p\big(w_{ij}^p x_i\big) \tag{14}$$

where $N$ is the number of input neurons in the layer. The value $z_j$ is then passed through a ReLU activation function resulting in the output $y_j = \max(0, z_j)$. Similarly, in convolutional layers, the additional synaptic dimension is incorporated into each filter. As a result, the summation in Eq. (14) also spans spatial locations and input channels.

For local synaptic plasticity modulation, we associate the local activity of each neuron with its output value. The eligibility trace is defined as follows:

$$\tilde{e}[t + 1] = \tilde{e}[t] - \frac{\tilde{e}[t]}{\tau} + z_j \tag{15}$$

where $\tau$ is a decay time constant. Each synaptic weight is updated using the following learning rule:

$$\Delta w_{ij}^p = -\eta \cdot \left| f_{\text{mod}}\big(\tilde{e}_{ij}\big) \right| \cdot \frac{\partial \mathcal{L}}{\partial w_{ij}^p} \tag{16}$$

where $\eta$ is the learning rate. The eligibility trace $\tilde{e}_{ij}$ keeps track of local synaptic activity. The function $f_{\text{mod}}$ uses this information to smoothly adjust the synaptic strength—potentiation or depression—similar to how synaptic plasticity is modulated in spiking neural models. When $P = 1$, Eq. (12)–Eq. (16) become the same as in standard ANNs, so this method remains fully compatible with existing implementations.

## A.2 Implementation Details

### A.2.1 Datasets.

The datasets used in our experiments are summarized in Table 4. We evaluate our approach on four standard continual learning benchmarks: Permuted MNIST (PMNIST), CIFAR-100, TinyImageNet, and 5-Datasets, which are presented in roughly increasing order of dataset complexity.

**PMNIST** is a variant of the original MNIST dataset consisting of $28 \times 28$ grayscale images of handwritten digits. Each task applies a fixed but unique random permutation to the pixel positions, making it a widely adopted benchmark for evaluating robustness in task-incremental learning. For PMNIST, we assign each task 60,000 training and 10,000 testing samples to increase the challenge.

**CIFAR-100.** CIFAR-100 is an object recognition dataset with 100 natural image classes. Following the protocol in Rebuffi et al. (2017), we partition the dataset into 10 tasks, each comprising 10 disjoint classes with their corresponding images.

**TinyImageNet** contains 100,000 $64 \times 64$ color images across 200 classes. We construct 40 sequential tasks by splitting the dataset into 5-way classification problems. For fair comparison, we randomly sample a subset of the original dataset and align the test set with the training set as in Serra et al. (2018).

**5-Datasets** combines tasks from five diverse datasets: MNIST, SVHN, Fashion-MNIST, CIFAR-10, and NotMNIST, each treated as an independent task. This setting evaluates the model's generalizability under distribution shift and cross-domain learning.

For CIFAR-100 and TinyImageNet, we follow standard settings with 500 training and 100/50 test images per class, respectively.

Table 4: Dataset statistics

| Dataset | PMNIST | CIFAR-100 | TinyImageNet | 5-Datasets |
|---|---|---|---|---|
| Tasks | 10 | 10 | 40 | 5 |
| Classes | 10 | 100 | 200 | / |
| Training Samples | 60,000 | 50,000 | 100,000 | / |
| Test Samples | 10,000 | 10,000 | 10,000 | / |

### A.2.2 EVALUATION METRICS

Following Kang et al. (2022a); Wortsman et al. (2020); Mallya et al. (2018), we evaluate all methods based on the following metrics:

*Accuracy (ACC)* measures the average of the final classification accuracy on all tasks:

$$\text{ACC} = \frac{1}{T} \sum_{i=1}^{T} acc_{T,i} \tag{17}$$

where $acc_{T,i}$ is the test accuracy for task $i$ after training on task $T$.

*Backward Transfer (BWT)* measures the influence of learning new tasks on the performance of previously learned ones. A negative BWT indicates forgetting, whereas a positive BWT suggests that learning later tasks improved the performance of earlier ones. BWT is computed as:

$$\text{BWT} = \frac{1}{T-1} \sum_{i=1}^{T-1} \left( acc_{T,i} - acc_{i,i} \right) \tag{18}$$

where $acc_{i,i}$ denotes the accuracy of task $i$ immediately after it is learned. A BWT close to zero implies stability, while highly negative values indicate catastrophic forgetting.

*Capacity (CAP)* measures the amount of network capacity used under each parameter pruning method Kang et al. (2022a). It accounts for both the proportion of trainable parameters and the efficiency of binary encoding. The CAP metric is defined as:

$$\text{CAP} = (1 - C) + \frac{(1-\alpha)N}{32} \tag{19}$$

where $\alpha$ is the average mask compression rate ($\alpha$=0.78), $N$ is the number of tasks, and $C$ is the percentage of non-fixed parameters. A smaller CAP value indicates higher effective network capacity.

*Average Order-normalized Performance Disparity (AOPD)* measures the robustness of these algorithms under different task orders. Following the protocol of Yoon et al. (2019), we assessed the task order robustness with the Order-normalized Performance Disparity (OPD) metric, which is computed as the disparity between the performance $\bar{A}_t$ of task $t$ on $R$ different task orders: $\text{OPD}_t \triangleq \max\{\bar{A}_t^1, \ldots, \bar{A}_t^R\} - \min\{\bar{A}_t^1, \ldots, \bar{A}_t^R\}$. The average OPD (AOPD) is defined by

$$\text{AOPD} \triangleq \frac{1}{T} \sum_{t=0}^{T-1} \text{OPD}_t \tag{20}$$

Table 5: Performance deviations of the proposed method and baselines on four datasets.

(a) PMNIST and 10-split CIFAR-100

| Network | Method | PMNIST | | 10-split CIFAR-100 | |
|---|---|---|---|---|---|
| | | ACC (%) | BWT (%) | ACC (%) | BWT (%) |
| SNN | MTL | 0.12 | / | 0.31 | / |
| | EWCKirkpatrick et al. (2017) | 0.51 | 0.42 | 0.63 | 0.57 |
| | HAT Serra et al. (2018) | 0.24 | 0.0 | 0.37 | 0.19 |
| | GPM Saha et al. (2021) | 0.43 | 0.34 | 0.41 | 0.38 |
| | HLOP Xiao et al. (2024) | 0.39 | 0.21 | 0.35 | 0.18 |
| | **MSCN** | **0.22** | **0.0** | **0.25** | **0.0** |
| ANN | MTL | 0.14 | / | 0.21 | / |
| | EWCKirkpatrick et al. (2017) | 0.56 | 0.01 | 0.57 | 0.49 |
| | GPM Saha et al. (2021) | 0.07 | 0.01 | 0.48 | 0.39 |
| | PackNet Mallya et al. (2018) | 0.04 | 0.0 | 0.41 | 0.0 |
| | SupSup Wortsman et al. (2020) | 0.09 | 0.0 | 0.32 | 0.0 |
| | WSN Kang et al. (2022a) | 0.07 | 0.0 | 0.29 | 0.0 |
| | TAMiL Bhat et al. (2023) | 0.17 | 0.04 | 0.36 | 0.49 |
| | **MSCN** | **0.19** | **0.0** | **0.23** | **0.0** |

(b) TinyImageNet and 5-Datasets

| Network | Method | TinyImageNet | | 5-Datasets | |
|---|---|---|---|---|---|
| | | ACC (%) | BWT (%) | ACC (%) | BWT (%) |
| SNN | MTL | 0.29 | / | 0.26 | / |
| | EWCKirkpatrick et al. (2017) | 0.59 | 0.72 | 0.48 | 0.65 |
| | HAT Serra et al. (2018) | 0.61 | 0.44 | 0.33 | 0.51 |
| | GPM Saha et al. (2021) | 0.46 | 0.29 | 0.36 | 0.42 |
| | HLOP Xiao et al. (2024) | 0.41 | 0.23 | 0.30 | 0.25 |
| | **MSCN** | **0.31** | **0.0** | **0.29** | **0.0** |
| ANN | MTL | 0.33 | / | 0.27 | / |
| | EWCKirkpatrick et al. (2017) | 0.44 | 0.03 | 0.26 | 0.02 |
| | GPM Saha et al. (2021) | 0.42 | 0.31 | 0.20 | 0.01 |
| | PackNet Mallya et al. (2018) | 0.35 | 0.0 | 0.12 | 0.0 |
| | SupSup Wortsman et al. (2020) | 0.40 | 0.0 | 0.21 | 0.0 |
| | WSN Kang et al. (2022a) | 0.34 | 0.0 | 0.13 | 0.0 |
| | TAMiL Bhat et al. (2023) | 0.31 | 0.13 | 0.22 | 0.08 |
| | **MSCN** | **0.27** | **0.0** | **0.21** | **0.0** |

### A.2.3 EXPERIMENT SETTINGS

All experiments were conducted on a Linux server equipped with an Intel Xeon Gold 5220 (2.20 GHz) CPU and two NVIDIA Tesla V100-SXM2 GPUs (32 GB each, driver 535.129.03). Following Kang et al. (2022a); Wortsman et al. (2020); Mallya et al. (2018), we use a two-layered MLP with 100 neurons per layer for PMNIST and use a modified version of AlexNet for the CIFAR-100 Split dataset and a reduced ResNet-18 Chaudhry et al. (2019); Saha et al. (2021) for 5-Datasets. For TinyImageNet,

we also use the same network architecture Gupta et al. (2020); Deng et al. (2021), which consists of 4 Conv layers and 3 fully connected layers. For a fair comparison, we follow the experimental setting in Kang et al. (2022a); Thapa & Li (2024), and all methods are evaluated under the same multi-head setting with known task labels. The hyperparameter settings are presented in Table 7.

Table 7: Experiment settings and hyperparameter configurations for different datasets

| Dataset | PMNIST | 10-split CIFAR-100 | TinyImageNet | 5-Datasets |
|---|---|---|---|---|
| learning rate | $1 \times 10^{-3}$ | $1 \times 10^{-3}$ | $1 \times 10^{-3}$ | $1 \times 10^{-3}$ |
| dropout rate | 0.2 | 0.2 | 0.5 | 0.2 |
| epochs | 5 | 200 | 10 | 100 |
| batch size | 10 | 64 | 10 | 64 |
| warmup ratio | 0.05 | 0.05 | 0.05 | 0.05 |
| optimizer | AdamW | AdamW | AdamW | AdamW |
| weight decay | $1 \times 10^{-2}$ | $1 \times 10^{-2}$ | $1 \times 10^{-2}$ | $1 \times 10^{-2}$ |
| architecture | Two-layered MLP | Modified AlexNet | 4 Conv layers and 3 Fully connected layers | Reduced ResNet18 |

### A.2.4 ARCHITECTURAL DETAILS

**Two-layered MLP:** In conducting the PMNIST experiments, we are following the exact setup as denoted by Saha et al. (2021) fully-connected network with two hidden layers of 100 neurons Lopez-Paz & Ranzato (2017).

**Modified AlexNet:** For the split CIFAR-100 dataset, we use a modified version of AlexNet similar to Joseph & Gu (2021); Saha et al. (2021).

**4 Conv layers and 3 Fully connected layers:** For TinyImageNet, we use the same network architecture as Joseph & Gu (2021); Deng et al. (2021).

**Reduced ResNet18:** In conducting the 5-Dataset experiments, we use a smaller version of ResNet18 with three times fewer feature maps across all layers as denoted by Lopez-Paz & Ranzato (2017).

### A.2.5 LIST OF MAIN NOTATIONS

In Table 8, we list the main notations used in this paper, together with brief explanations, enabling quick reference to the meaning of each symbol.

### A.3 EXTRA EXPERIMENTS

### A.3.1 INTEGRATION WITH DIFFERENT TYPES OF METHODS

To further evaluate the applicability of our MSCN, we integrated it with different types of methods and conducted additional experiments on both 10-split CIFAR-100 and TinyImageNet under the same parameter budget. As shown in Table 9, integrating MSCN into regularization-based and replay-based methods consistently improves BWT. Notably, combining MSCN with ER on TinyImageNet improves BWT by 53.8% (an absolute decrease of 10.69). The observed BWT improvements are attributed to the high capacity efficiency of MSCN, which arises from the independent optimization of multiple parallel synapses, as demonstrated in Fig. 7. Such higher capacity efficiency has been shown to reduce catastrophic forgetting Hung et al. (2019); Mirzadeh et al. (2022); Farajtabar et al. (2020). Meanwhile, the modulation mechanism further enhances this property by depressing the effect of noisy samples and strengthening learning on clean ones. In contrast, for architecture-based methods, BWT remains zero because the weights of past tasks are frozen, which is exactly as expected. At the same time, when our MSCN is incorporated, all three types of methods achieve improved accuracy. These additional experiments further highlight the robustness of our MSCN.

### A.3.2 LAYER-WISE CAPACITY ANALYSIS

As shown in Fig. 8, we analyze the synaptic capacity usage across three benchmarks: PMNIST, CIFAR-100 Split, and TinyImageNet. For each dataset, we measure the percentage of utilized synapses in each layer as tasks are incrementally learned. Across all three datasets, we observe a consistent pattern: capacity usage increases rapidly during the initial tasks, then gradually slows

Table 8: List of main notations

| Notation | Description |
|---|---|
| $m_j$ | Binary mask selecting active synapses for task $j$ |
| $M_{j-1}$ | Accumulated mask of all previous tasks up to $j-1$ |
| $r$ | Learnable relevance score for each synapse |
| $c$ | Layer-wise capacity ratio for subnet selection |
| $P$ | Number of parallel synapses in each connection (synapse count) |
| $w_{ip}$ | Weight of the $p$-th synapse from presynaptic neuron $i$ |
| $N$ | Number of presynaptic neurons |
| $V(t)$ | Membrane potential of a spiking neuron at time $t$ |
| $\tau_m$ | Membrane time constant in LIF neurons |
| $V_{\text{rest}}$ | Resting potential of the spiking neuron |
| $I(t)$ | Total synaptic input current at time $t$ |
| $\text{PSP}_{ip}(t)$ | Postsynaptic potential from synapse $p$ of neuron $i$ |
| $\tau_{s,ip}$ | Decay constant of the $p$-th parallel synapse of neuron $i$ |
| $K_{ip}(t)$ | Synaptic kernel of the $p$-th synapse of neuron $i$ |
| $t_{ip}^f$ | Arrival time of the $f$-th spike at synapse $(i, p)$ |
| $\tilde{e}$ | Eligibility trace representing local synaptic activity |
| $\tau$ | Decay time constant of the eligibility trace |
| $f_{\text{mod}}(\tilde{e})$ | Modulation function for $\tilde{e}$ |

Table 9: Integration with different types of methods.

| Type | Method | 10-split CIFAR-100 | | TinyImageNet | |
|---|---|---|---|---|---|
| | | ACC (%) ↑ | BWT (%) ↑ | ACC (%) ↑ | BWT (%) ↑ |
| regularization-based | EWC | 72.77 ($\pm 0.57$) | -3.59 ($\pm 0.49$) | 64.51 ($\pm 0.44$) | -0.04 ($\pm 0.03$) |
| | **EWC+MSCN** | **73.26** ($\pm 0.66$) | **-2.78** ($\pm 0.19$) | **64.98** ($\pm 0.54$) | **-0.03** ($\pm 0.01$) |
| replay-based | ER | 70.07 ($\pm 0.73$) | -7.70 ($\pm 0.59$) | 48.32 ($\pm 0.91$) | -19.86 ($\pm 0.70$) |
| | **ER+MSCN** | **71.13** ($\pm 0.62$) | **-5.24** ($\pm 0.51$) | **49.26** ($\pm 0.84$) | **-9.17** ($\pm 0.55$) |
| architecture-based | Bayesian | 75.57 ($\pm 0.38$) | 0.00 ($\pm 0.00$) | 73.93 ($\pm 0.36$) | 0.00 ($\pm 0.00$) |
| | **Bayesian+MSCN** | **76.48** ($\pm 0.34$) | **0.00** ($\pm 0.00$) | **74.56** ($\pm 0.33$) | **0.00** ($\pm 0.00$) |

down as more tasks are introduced. This effect is particularly pronounced in the fully connected layers, such as FC1, which tend to accumulate more synaptic updates compared to early convolutional layers. The underlying reason is that the model needs to allocate new synaptic resources to encode novel task-specific features at the beginning. However, as training progresses, many new tasks can be handled by reusing synapses that represent similar features, reducing the need for additional capacity. This confirms the model's ability to reuse past representations more effectively as it acquires more knowledge, leading to a slower growth in capacity usage over time.

### A.3.3 ANALYSIS OF SYNAPSE COUNT

Fig. 9 illustrates the relationship between synapse count (denoted as $P$) and average accuracy as the number of tasks increases, evaluated on PMNIST and CIFAR-100 Split. We vary the number of synapses per connection across five settings ($P$=2, 4, 6, 8, 10) and track model performance throughout the incremental learning process. We observe that on PMNIST, accuracy remains high across all configurations; however, larger synapse counts (e.g., $P$=8, 10) tend to deliver more stable performance over multiple tasks. On CIFAR-100 Split, the benefits of increased synaptic capacity become more evident: higher $P$ values consistently result in better average accuracy, particularly as the number of tasks grows. These results confirm that synaptic multiplicity enhances the model's ability to retain knowledge and generalize over longer task sequences.

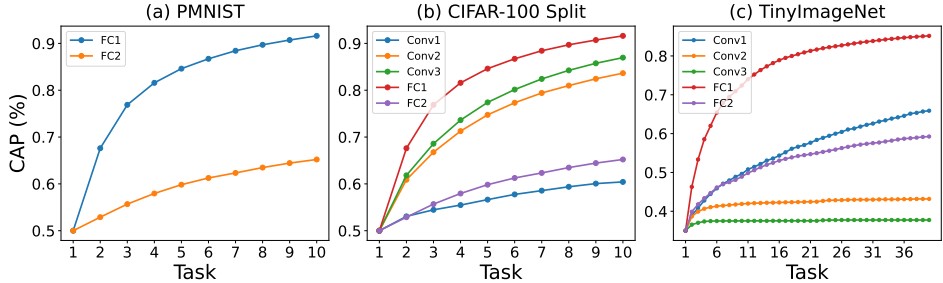

Figure 8: Synaptic capacity usage as the number of tasks increases on three benchmarks. Each curve shows the percentage of active synapses per layer as tasks are incrementally introduced.

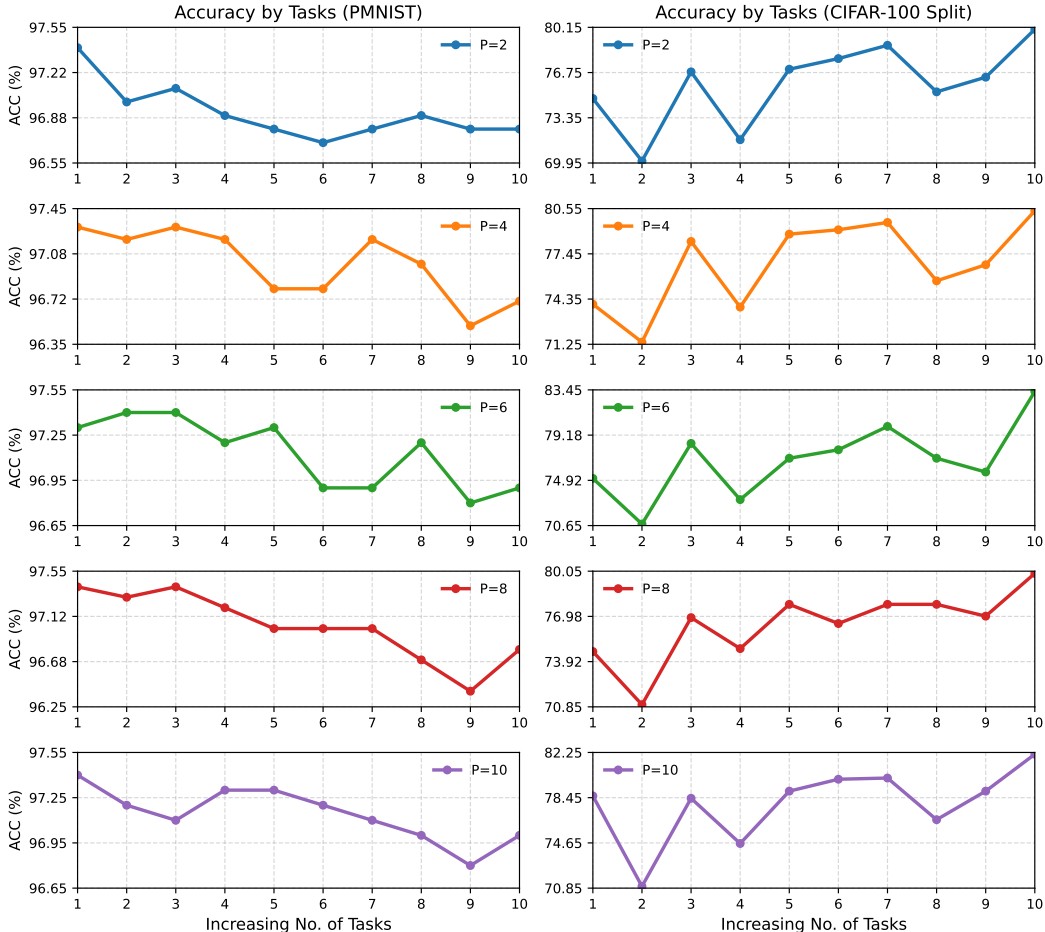

Figure 9: Average accuracy with increasing number of tasks under different synapse counts.

### A.3.4 ORDER ROBUSTNESS ANALYSIS

To further investigate the robustness of our method to task permutations, we conduct additional experiments on the CIFAR-100 Split benchmark using five randomly shuffled task orders. Fig. 10 presents the per-task accuracy across all 10 tasks for three representative baselines—EWC Kirkpatrick et al. (2017), GPM Saha et al. (2021), and WSN Kang et al. (2022a)—alongside our proposed MSCN. We observe that EWC (Fig. 10a) and GPM (Fig. 10b) are highly sensitive to task order, exhibiting considerable variance in accuracy for the same task index across different permutations. In contrast, WSN (Fig. 10c) achieves more stable performance, though moderate fluctuations persist, particularly

on later tasks. Notably, our method, MSCN (Fig. 10d), maintains consistently high accuracy across all permutations and task indices, with significantly reduced inter-order variance. These results show that MSCN is robust to task order, ensuring stability in dynamic environments.

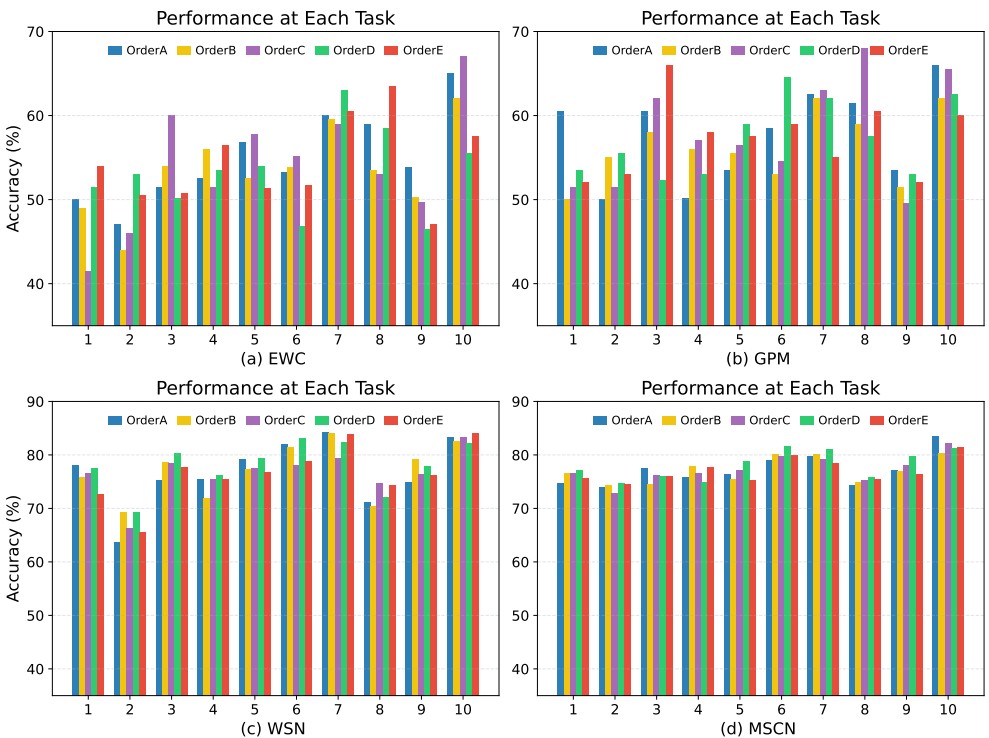

Figure 10: Task order robustness comparison on CIFAR-100 Split. Bar plots show per-task accuracy under five different task sequences.

