# OpenReview forum: "Multi-Synaptic Cooperation: A Bio-Inspired Framework for Robust and Scalable  Continual Learning"
_ICLR.cc/2026/Conference — ICLR 2026 Poster_

### Official Review · Reviewer_M3Db · 2025-10-19

**Soundness:** 3
**Presentation:** 3
**Contribution:** 3
**Rating:** 6
**Confidence:** 4

**Summary:**

This paper proposes the Multi-Synaptic Cooperative Network (MSCN), a biologically inspired continual learning framework that models cooperative interactions among multiple synapses.The authors draw inspiration from multi-synaptic connectivity in biological neurons and local synaptic plasticity modulated by eligibility traces. Each neural connection is modeled as having multiple parallel synapses, whose plasticity is modulated locally based on activity. This enables adaptive task-dependent plasticity, allowing the model to selectively activate or inhibit synapses during new task learning while minimizing interference with prior knowledge. Extensive experiments across both spiking and non-spiking architectures on four benchmark datasets demonstrate that MSCN consistently outperforms state-of-the-art methods in both accuracy and robustness.

**Strengths:**

1. This paper is well-written and easy to follow.

2. The proposed Multi-Synaptic Cooperative Network (MSCN) provides an innovative, biologically-grounded strategy for expanding network capacity in continual learning.

3. The proposed method applies to both spiking and non-spiking neural networks, and achieves 0 BWT.

**Weaknesses:**

1. The experiments are mainly performed with CIFAR-100 and Tiny-ImageNet of relatively small image scales. Does the proposed method apply to larger-scale images, such as 224*224 images of ImageNet (subsets)?

2. The proposed method seems to provide very marginal performance gains under more clear distribution shifts (i.e., 5-Dataset).

3. As an architecture-based method, similar to other baselines, the proposed method may apply to only task-incremental setting.

4. What’s the total parameter cost of the proposed method, compared to other baselines?

**Questions:**

My major concerns lie in the applicability of the proposed method. Please refer to the Weaknesses.

---

> ### Author Response · Authors · 2025-11-21
>
> We sincerely appreciate the reviewer's recognition of the novelty of our method and the clarity of our writing, and we are deeply grateful for your detailed and constructive comments.
>
> >**Weaknesses:**
>
> >**W1: Apply to larger-scale images**
>
> We follow common practice in continual learning and evaluate MSCN on standard benchmark datasets such as Permuted MNIST, CIFAR-100 Split, TinyImageNet, and 5-Datasets, which are also widely used by prior methods [1,2,3,4]. This **allows a fair and consistent comparison with existing work.**
> We agree that experiments on larger-scale datasets would provide a better assessment of the practical scalability of MSCN. Due to time limitations, we were unable to conduct experiments on ImageNet-1K. We are excited to further investigate MSCN on larger-scale benchmarks in future work to gain a deeper understanding of its applicability. Nevertheless, we believe that the current results and analyses already provide strong support for the key contributions and claims presented in our paper. Thanks.
>
> >**W2: The proposed method seems to provide very marginal performance gains under more clear distribution shifts (i.e., 5-Dataset).**
>
> Thank you for this careful observation. As indicated in the title, **our work primarily focuses on the scalability and robustness rather than accuracy**. Across all four datasets, our MSCN consistently achieves the shortest training time and the lowest AOPD (the best robustness), and the results in Figure 7 further demonstrate its superior capacity efficiency.
> In practice, we observe that the relative accuracies of all methods vary across different datasets. For a more comprehensive assessment of generalization, we therefore analyzed the average training time, accuracy, and AOPD across all four datasets in Table 3. The results are summarized in the table below.
>
> Average method performance across PMNIST, 10-split CIFAR-100, TinyImageNet, and 5-Datasets
> | Method  | Average Training Time ↓ | Average ACC ↑ | Average AOPD ↓ |
> |:-:|:-:|:-:|:-:|
> | PackNet | 1.58 ± 0.11| 79.32 ± 0.19 | 4.62 |
> | SupSup  | 1.41 ± 0.09 | 81.22 ± 0.20 | 3.60 |
> | WSN | 1.28 ± 0.06 | 84.62 ± 0.25  | 2.31|
> | **MSCN**|  **1.14±0.06**| **85.17±0.16** | **2.16** |
>
> The experimental results indicate that our MSCN achieves the best balance among computational efficiency, ACC, and robustness. Therefore, although the accuracy varies on individual datasets, **MSCN always achieves the best overall performance.**
> >**W3 : As an architecture-based method, similar to other baselines, the proposed method may apply to only task-incremental setting.**
>
> >**Questions: My major concerns lie in the applicability of the proposed method.**
>
> As discussed in the Related Work section (lines 135–138), we implement our method in an architecture-based, task-incremental setting to better study its capacity efficiency and robustness, following the same experimental setup as recent works [3,4].
> At the same time, our method is **not restricted to any particular type of experimental setting or continual learning approach**. To further substantiate this point, we conducted additional experiments in different settings, combining MSCN with representative methods of various types and evaluating accuracy. The results are reported in the table below.
> | Method         | PMNIST           | 10-split CIFAR-100 | TinyImageNet      | 5-Datasets       |
> |:--------------|:-----------------|:------------------:|:-----------------:|:----------------:|
> | EWC [1]       | 92.01 ± 0.56     | 72.77 ± 0.57       | 64.51 ± 0.44      | 88.64 ± 0.26     |
> | **EWC+MSCN**      | **92.87 ± 0.18**     | **73.26 ± 0.66**       | **64.98 ± 0.54**      | **89.23 ± 0.41**     |
> | HLOP [3]      | 95.15 ± 0.39     | 75.58 ± 0.35       | 71.40 ± 0.41      | 88.65 ± 0.30     |
> | **HLOP+MSCN**     | **95.74 ± 0.31**     | **76.19 ± 0.52**       | **71.93 ± 0.69**      | **89.06 ± 0.29**     |
> | Bayesian [4]  | 96.74 ± 0.28     | 75.57 ± 0.38       | 73.93 ± 0.36      | 93.36 ± 0.31     |
> | **Bayesian+MSCN** | **97.51 ± 0.25**     | **76.48 ± 0.34**       | **74.56 ± 0.33**      | **94.07 ± 0.27**     |
>
> The experiment results show that combining each of them with our MSCN mechanism consistently improves accuracy, demonstrating the applicability of MSCN and indicating that its effectiveness is not limited to a specific experimental setting or continual learning method, e.g. architecture-based.

---

> ### Author Response · Authors · 2025-11-21
>
> >**W4: What’s the total parameter cost of the proposed method, compared to other baselines?**
>
> As shown in Table 3, all results are reported under the same parameter budget. The detailed parameter cost for each dataset and architecture is summarized in the table below.
> | Dataset          | Architecture        | #Params (M) |
> |:----------------|:--------------------|:-----------:|
> | PMNIST          | 2-Layer MLP         | 0.19M       |
> | CIFAR-100 Split | Improved AlexNet    | 13.20M      |
> | TinyImageNet    | TinyNet             | 5.62M       |
> | 5-Datasets      | Reduced ResNet-18   | 2.21M       |
>
>
> **References:**
>
> [1] Kang H, Mina R J L, Madjid S R H, et al. Forget-free continual learning with winning subnetworks[C]//International conference on machine learning. PMLR, 2022: 10734-10750.
>
> [2] Wortsman M, Ramanujan V, Liu R, et al. Supermasks in superposition[J]. Advances in neural information processing systems, 2020, 33: 15173-15184.
>
> [3] Xiao M, Meng Q, Zhang Z, et al. Hebbian learning based orthogonal projection for continual learning of spiking neural networks[J]. arXiv preprint arXiv:2402.11984, 2024.
>
> [4] Thapa J, Li R. Bayesian adaptation of network depth and width for continual learning[C]//International Conference on Machine Learning. PMLR, 2024, 235: 48038-48061.

---

> ### Comment · Reviewer_M3Db · 2025-11-23
>
> I thank the authors for their rebuttal, which addressed some of my concerns. I therefore keep my original rating and cannot be more positive at the current stage.

---

> ### Author Response · Authors · 2025-11-24
>
> Dear Reviewer M3Db,
>
> We thank you for your response and for acknowledging that several of your concerns were addressed. While we fully respect your decision to maintain the current positive rating at this stage, we would like to share additional **new and exciting** results that we conducted after the first-round rebuttal, following Reviewer jVm8’s request for explicit BWT evaluation.
>
> To provide a more comprehensive assessment of our MSCN across different continual learning paradigms, we performed new experiments on both 10-split CIFAR-100 and TinyImageNet under the same parameter budget for regularization, replay, and architecture-based methods. These extended evaluations reveal that MSCN not only maintains **the previously reported accuracy advantages** but also achieves **substantial improvements in BWT for both regularization and replay-based methods,** further strengthening the empirical evidence for MSCN’s robustness, generality, and reduced interference during sequential learning.
>
> We hope that these additional results—motivated directly by the review process—offer a clearer and more complete picture of the strengths and maturity of our approach. **If you find these new findings helpful and convincing, we would be sincerely grateful for any reconsideration of the current rating.** We deeply appreciate your time, constructive feedback, and thoughtful evaluation of our work.
>
>  For your convenience, the newly added results and accompanying explanations are also included below:
>
> | Type                 | Method            | 10-split CIFAR-100 ACC ↑ | 10-split CIFAR-100 BWT ↑ | TinyImageNet ACC ↑ | TinyImageNet BWT ↑ |
> | -------------------- | ----------------- | ------------------------ | ------------------------ | ------------------ | ------------------ |
> | regularization-based | EWC               | 72.77 ± 0.57             | -3.59±0.49               | 64.51 ± 0.44       | -0.04±0.03         |
> | regularization-based | **EWC+MSCN**      | **73.26 ± 0.66**         | **-2.78±0.19**           | **64.98 ± 0.54**   | **-0.03±0.01**     |
> | replay-based         | ER                | 70.07 ± 0.73             | -7.70±0.59               | 48.32 ± 0.91       | -19.86±0.70        |
> | replay-based         | **ER+MSCN**       | **71.13 ± 0.62**         | **-5.24±0.51**           | **49.26 ± 0.84**   | **-9.17±0.55**     |
> | architecture-based   | Bayesian          | 75.57 ± 0.38             | 0.00±0.00                | 73.93 ± 0.36       | 0.00±0.00          |
> | architecture-based   | **Bayesian+MSCN** | **76.48 ± 0.34**         | **0.00±0.00**            | **74.56 ± 0.33**   | **0.00±0.00**      |
>
> The results show that, **when incorporating MSCN into regularization-based and replay-based methods, it almost always improves BWT**. Notably, combining our MSCN with ER on TinyImageNet improves BWT by 53.8% (an absolute decrease of 10.69).
>
> The observed BWT improvements are due to the high capacity efficiency of our MSCN, which **arises from the independent optimization of multiple parallel synapses**, as demonstrated in Fig. 7. Such higher capacity efficiency has been shown to reduce catastrophic forgetting [1,2,3]. Meanwhile, the modulation mechanism further enhances this property by **depressing the effect of noisy samples and strengthening learning on clean ones**. In contrast, for architecture-based methods, BWT remains zero because the weights of past tasks are frozen, and this result is exactly as expected. At the same time, when our MSCN is incorporated, all three types of methods achieve improved accuracy.
>
> In summary, these additional experiments, conducted in direct response to Reviewer jVm8, further strengthen our results and highlight the robustness of our MSCN approach. We hope this will further highlight the contributions of our work.
>
> **References:**
>
> [1] Hung C Y, Tu C H, Wu C E, et al. Compacting, picking and growing for unforgetting continual learning[J]. Advances in neural information processing systems, 2019, 32.
>
> [2] Mirzadeh S I, Chaudhry A, Yin D, et al. Wide neural networks forget less catastrophically[C]//International conference on machine learning. PMLR, 2022: 15699-15717.
>
> [3] Farajtabar M, Azizan N, Mott A, et al. Orthogonal gradient descent for continual learning[C]//International conference on artificial intelligence and statistics. PMLR, 2020: 3762-3773.

---

### Official Review · Reviewer_dFvv · 2025-10-29

**Soundness:** 2
**Presentation:** 2
**Contribution:** 2
**Rating:** 4
**Confidence:** 4

**Summary:**

This paper mainly focuses on the catastrophic forgetting in continual learning and presents a biologically inspired framework termed the Multi-Synaptic Cooperative Network (MSCN). Drawing inspiration from the organization of biological neural systems, MSCN enhances the representational capacity of a fixed network architecture by associating multiple plastic synapses with each neuronal connection, thereby enabling flexible information encoding without structural expansion. In addition, the authors propose a bio-inspired plasticity modulation mechanism that employs local eligibility traces to adaptively regulate synaptic updates. During learning, this mechanism selectively strengthens, weakens, or maintains specific synapses based on their relevance to the current task. This strategy helps prevent interference between tasks and reduces forgetting. Extensive experiments on multiple benchmark datasets demonstrate the superior performance and robustness of MSCN compared to existing approaches

**Strengths:**

1.Sound methodology: The proposed MSCN framework is well-motivated by biological inspiration and conceptually well-grounded. The idea of equipping each neuronal connection with multiple synapses to enhance representational capacity within a fixed architecture is both intuitively appealing and theoretically sound.
2.Comprehensive experiments: The experimental evaluation is thorough and well-executed. The paper benchmarks MSCN on multiple datasets using both spiking neural networks (SNNs) and artificial neural networks (ANNs), demonstrating its broad applicability across architectures. Moreover, the analysis under various experimental settings—such as different task order scenarios—provides a comprehensive understanding of the method’s robustness and generalization ability.

**Weaknesses:**

1. Lack of clarity: Although the overall framework is well-structured, the definition and notation of the multiple-synapse mechanism are somewhat ambiguous. In particular, the variables such as s, j, and f in Equations (2) and (3) are not clearly explained.
2. Limited performance improvement and trade-offs: The performance gains of MSCN are relatively modest, particularly on the 5-Datasets benchmark. Moreover, as shown in Table 3, although MSCN achieves shorter training times, this advantage comes at the cost of reduced performance, suggesting a trade-off between efficiency and accuracy that warrants further discussion.
3. Scalability and applicability to large-scale settings: The current experiments are limited to datasets with relatively small numbers of classes and models of modest size (e.g., MLPs, ResNet-18). The absence of evaluations on large-scale datasets such as ImageNet-1K or on modern architectures like Vision Transformers (ViT) makes it difficult to assess MSCN’s scalability and practicality in more complex, real-world scenarios. In particular, the computational and memory overhead, as well as the training stability of multi-synaptic connectivity in large models, remain unexplored.

**Questions:**

1.In Figure 3 (a–c), the current grayscale visualization makes it somewhat difficult to distinguish key patterns. Using color plots or more distinctive contrast could improve readability and help highlight the differences between conditions more clearly.
2. Moreover, in Figure 3, EWC shows large accuracy fluctuations on the first task, likely due to task difficulty variations, whereas MSCN maintains notably stable performance. It would be helpful to clarify how MSCN achieves such robustness to task order and difficulty.
3.Most baseline comparisons are made against architecture-based methods, yet the manuscript does not clarify the network size or total parameter count. Since each connection in MSCN contains K synapses, it is unclear how this differs in practice from simply increasing the network width by a factor of K. When K = 3, for instance, does the improvement primarily stem from the increased parameter count rather than the proposed mechanism itself?

---

> ### Author Response · Authors · 2025-11-21
>
> We appreciate  the reviewer's recognition of the soundness of our method and the comprehensiveness of our experimental design, and we are sincerely grateful for your detailed and constructive comments.
>
> >**Weaknesses:**
>
> >**W1: Improvement of clarity**
>
> Thanks for your careful comment. In the revised manuscript, we **provide more detailed explanations of the variables and add a notation table for better understanding.**
> For your convenience, the symbols you mentioned are explained as follows:
> - The index $i$ denotes the $i-th$ input neuron
> - $\mathrm{PSP}_{ip}(t)$ represents the change in the membrane potential of the LIF neuron at time $t$ induced by this input neuron through its $p-th $ synapse
> - The variable $f$ denotes the spike train, and $j$ indexes the spikes emitted by the LIF neuron, with $t_s^{j}$ indicating the time of its $j-th$ spike.
> Since a relatively large number of symbols are involved, we hope that the notation table newly added in Appendix A.2 will help clarify the notation and address your concern.
>
> >**W2: Limited performance improvement and trade-offs**
>
> As indicated in the manuscript title, our work primarily focuses on scalability and robustness rather than accuracy. Across all four datasets, MSCN consistently achieves the shortest training time and the lowest AOPD, and the results in Figure 7 further demonstrate its superior capacity efficiency.
> We agree that there is a trade-off between efficiency and accuracy, but **this phenomenon is a general characteristic of most machine learning methods [1,2] rather than a limitation specific to our approach.** To obtain a more comprehensive assessment of the trade-off, we therefore analyzed the average performance across all four datasets in Table 3. The results are summarized in the table below.
>
> Average method performance across PMNIST, 10-split CIFAR-100, TinyImageNet, and 5-Datasets
>
> | Method  | Average Training Time ↓ | Average ACC ↑ | Average AOPD ↓ |
> |:-:|:-:|:-:|:-:|
> | PackNet | 1.58 ± 0.11| 79.32 ± 0.19 | 4.62 |
> | SupSup  | 1.41 ± 0.09 | 81.22 ± 0.20 | 3.60 |
> | WSN | 1.28 ± 0.06 | 84.62 ± 0.25  | 2.31|
> | **MSCN**|  **1.14±0.06**| **85.17±0.16** | **2.16** |
>
> These results show that, under the same parameter budget, our MSCN attains the shortest average training time, the highest average accuracy (85.17%), and the lowest average AOPD (2.16). Although performance may vary on individual datasets, the average performance demonstrates that **MSCN achieves the best trade-off between efficiency and accuracy as compared to other baselines.**
>
> >**W3: Scalability and applicability to large-scale settings**
>
> We follow common practice and evaluate MSCN on benchmark datasets (e.g., Permuted MNIST, CIFAR-100 Split, TinyImageNet, and 5-Datasets) that are widely adopted by prior continual learning methods [3,4]. This **ensures fair and consistent comparisons within the existing literature.**
> Nevertheless, we agree that demonstrating effectiveness on larger-scale datasets and architectures is important for assessing the practical scalability of MSCN. Due to the limited review period, we were unable to conduct experiments on ImageNet-1K or ViT-based models. Given the simplicity and compatibility of MSCN, we are very excited to explore this direction in future work.
> Finally, we believe that the current results and analyses already provide strong support for the key contributions and claims presented in our paper. Thanks.

---

> ### Author Response · Authors · 2025-11-21
>
> >**Questions**
>
> >**Q1: Color of the figure**
>
> Thanks for your detailed suggestions. In the revised manuscript, we have redrawn the subplots in Figure 3(a–c). Specifically, each bar chart now adopts a higher-contrast color scheme, which makes the differences between the compared methods clearer.
>
> >**Q2: Clarify how MSCN achieves such robustness to task order and difficulty**
>
> The robustness gains of our method primarily arise **from the modulation mechanism that is driven by local synaptic activity.** This mechanism rescales the gradient at each synapse while preserving the update direction, thereby achieving a balance between stability and plasticity at the synaptic level.
> Specifically, when a task is noisy and induces excessively strong synaptic activity, the modulation reduces the magnitude of weight updates, which helps protect consolidated knowledge and limits excessive performance drift. When synaptic activity lies in a moderate range, the modulation allows larger updates, which strengthens learning of the current task. In this way, modulation **reduces the impact of high-noise tasks on overall performance**, reduces interference across tasks, and improves robustness to task order. In the revised manuscript, we have added a more detailed explanation of this modulation mechanism to address this concern.
>
> >**Q3.1: Clarify the network size or total parameter count**
>
> In the revised manuscript, we **provide a detailed description of the model size and its total number of parameters**. For each of the four datasets, we report the corresponding network architecture and its total parameter count, as summarized below.
> | Dataset          | Architecture        | #Params (M) |
> |:----------------|:--------------------|:-----------:|
> | PMNIST          | 2-Layer MLP         | 0.19M       |
> | CIFAR-100 Split | Improved AlexNet    | 13.20M      |
> | TinyImageNet    | TinyNet             | 5.62M       |
> | 5-Datasets      | Reduced ResNet-18   | 2.21M       |
>
> >**Q3.2: How MSCN differs in practice from simply increasing the network width**
>
> In Table 3, MSCN is implemented on top of the WSN [3] backbone. When we introduce $K=3$ parallel synapses, the corresponding WSN reference serves exactly as the baseline that simply increases the network width, as the reviewer suggested. This design ensures that all compared methods operate under the same parameter budget, allowing a fair and controlled comparison of performance gains.
> The results in Table 3 and our response to your W2 above show that, under the same parameter budget, our MSCN consistently achieves the best overall performance. This indicates that the performance gains of MSCN **do not result from an increased number of parameters, but from the proposed multi-synaptic cooperation  mechanism.**
>
>
> **References:**
>
> [1] Xu Z, Liu Z, Chen B, et al. Compress, then prompt: Improving accuracy-efficiency trade-off of llm inference with transferable prompt[J]. arXiv preprint arXiv:2305.11186, 2023.
>
> [2] Shengyuan C, Cai Y, Fang H, et al. Differentiable
> neuro-symbolic reasoning on large-scale knowledge graphs[J]. Advances in Neural Information Processing Systems, 2023, 36: 28139-28154.
>
> [3] Kang H, Mina R J L, Madjid S R H, et al. Forget-free continual learning with winning subnetworks[C]//International conference on machine learning. PMLR, 2022: 10734-10750.
>
> [4] Thapa J, Li R. Bayesian adaptation of network depth and width for continual learning[C]//International Conference on Machine Learning. PMLR, 2024, 235: 48038-48061.

---

### Official Review · Reviewer_CnT3 · 2025-10-31

**Soundness:** 3
**Presentation:** 2
**Contribution:** 2
**Rating:** 6
**Confidence:** 4

**Summary:**

This paper proposes the Multi-Synaptic Cooperative Network (MSCN), a biologically inspired framework for continual learning. The core idea is to model multiple synaptic connections between neuron pairs, modulated by local synaptic activity via eligibility traces, to enhance representational capacity and robustness within a fixed network architecture.

**Strengths:**

1. The idea of integrating multi-synaptic cooperation with localized plasticity modulation is biologically inspired and potentially novel.
2. The focus on improving robustness to task-order variation and scalability without dynamic network expansion targets significant practical limitations of existing architecture-based methods.

**Weaknesses:**

1. The abstract and introduction present the work as "the first continual learning framework that explicitly leverages the multi-synaptic collaboration mechanism." This claim is overstated and needs significant qualification. The concept of having multiple, plastic pathways or "synapses" between units is not new.
2. The authors must clearly differentiate MSCN from these related approaches. Is the novelty primarily in the specific biological inspiration(the explicit modeling of multiple contacts per axon-dendrite pair) and its integration with eligibility traces? This needs to be precisely stated and contrasted with the prior art.
3. The statement that "the brain achieves continual learning without suffering from dynamic structural growth" is a simplification. While the overall neuron count is relatively stable in adulthood, synaptic formation and pruning (structural plasticity) are fundamental to learning.
4. While capacity (CAP) is analyzed, the direct computational cost (FLOPs, memory footprint, training/inference time) of maintaining and updating P synapses per connection is not thoroughly discussed. For a method aiming for scalability, this is an important practical consideration. Table 3 mentions training time but under a fixed parameter budget; a discussion of the inherent cost of the multi-synaptic structure is warranted.

**Questions:**

1. How does the proposed multi-synaptic connectivity fundamentally differ from, for example, maintaining a per-parameter importance score (as in regularization-based methods) or learning a supermask over a fixed, over-parameterized network? What specific capabilities does the explicit modeling of multiple, independent synapsesenable that these other approaches do not?
2. The eligibility trace mechanism is a key component. In the context of ANNs trained with backpropagation, what is the precise advantage of this bio-inspired, activity-dependent modulation over a standard gradient-based update? The ablation shows it helps, but the theoretical or empirical reasonwhy this specific form of modulation is particularly effective for task-order robustness remains unclear.
3. The results show performance saturates with increasing P. Was there any exploration into making P adaptive or variable across layers/connections? The biological system it aims to emulate is highly heterogeneous, not uniform.
4. Have you considered comparing against a strong, non-biologically-inspired baseline that also uses a fixed but over-parameterized network (e.g., a simple, wider network) to see if the gains are specifically due to the structured, multi-synapticnature of the over-parameterization, rather than just increased capacity?

---

> ### Author Response · Authors · 2025-11-21
>
> We sincerely appreciate your recognition of the novelty and clarity of our work and your insightful comments, and we hope the following responses can address your concerns.
>
> **Weaknesses:**
> >**W1: Claim needs qualification**
>
> We agree that prior work has already explored multi-synaptic connections, as we also have discussed in the Related Work section of our initial submitted version (lines 148–149).
> We would like to emphasize that our contribution is **not simply the adoption of multi-synaptic connections**. Instead, we explicitly propose and implement a **multi-synaptic cooperation** mechanism for continual learning, which **combines a multi-synaptic connectivity structure with a plasticity modulation mechanism** based on local synaptic activity to improve scalability and robustness to task order. To avoid any ambiguity, we have revised the manuscript to make this scope more explicit.
>
> >**W2: Clearly differentiate MSCN from related approaches**
>
> Our main contribution lies in the proposed multi-synaptic cooperation mechanism. Compared with prior art, our approach differs in several key aspects:
> - Existing work on multi-synapse connectivity generally treats multiple contacts independently [1, 2, 3]. In contrast, our MSCN explicitly **introduces a plasticity modulation mechanism driven by local synaptic activity and integrates it with a multi-synaptic connectivity structure.**
> - Our MSCN is designed to enhance the scalability and task-order robustness of continual learning methods. These **goals differ from those of prior approaches** such as [1, 2], which primarily aim to improve accuracy.
> Following your suggestion, we have revised the Related Work section to state this novelty more precisely and to more clearly contrast MSCN with representative multi-synapse methods.
>
> >**W3: The statement that "the brain achieves continual learning without suffering from dynamic structural growth" is a simplification.**
>
> We appreciate this careful observation. The original phrasing was a space-limited simplification intended to emphasize the relative stability of adult neuron counts, not to deny structural plasticity. In the revised manuscript, we have added a clarification to explicitly highlight the importance of synaptic formation and pruning.
>
> >**W4: The direct computational cost needs to be thoroughly discussed**
>
> We would like to emphasize that our method is not merely a multi-synaptic structure, but a combination with the local plasticity modulation mechanism. **Table 3 is designed to evaluate the performance of the multi-synaptic mechanism under the same parameter budget**, rather than to analyze its inherent computational cost. To address the reviewer's concern, **we have additionally conducted an ablation study on computational cost**, where we report training time and peak GPU memory to quantify the overhead introduced by each component of the multi-synaptic cooperation mechanism. The results are presented in the table below. Unlike the same parameter budget setting in Table 3, the following experiments evaluate both the vanilla models and the models combined with our multi-synaptic cooperation mechanism.
>
> | Multi Synapse | Modulation | Training Time (hours) ↓ | Peak GPU Memory (MB) ↓ | ACC (%) ↑| AOPD ↓ |
> |:-:|:-:|:-:|:-:|:-:|:-:|
> | ✓  | ✓  | 0.65 | 3762 | 77.37 (± 0.23) | 2.41 |
> | ✗ | ✓ | 0.39  | 2466 | 77.03 (± 0.22) | 3.74 |
> | ✓ | ✗ | 0.59 | 3584 |76.81 (± 0.25) | 4.62  |
> | ✗ | ✗ | 0.37 | 2300 | 76.34 (± 0.24) | 6.31  |
>
> The experimental results show that, with the introduction of the multi-synaptic structure, MSCN incurs a moderate increase in computational cost. The result of our MSCN in Table 3 also shows that, in the same parameter budget, the **additional overhead introduced by our modulation mechanism is negligible**. Given the corresponding and substantial improvements in ACC and AOPD, particularly the reduction of AOPD from 6.31 to 2.41, together with the potential of the multi-synaptic cooperation mechanism, we regard this **extra inherent computational cost as a reasonable trade-off** at this early stage. **This complexity is expected and biologically plausible**, as multi-synaptic connections are widely observed in the brain [2,3].

---

> ### Author Response · Authors · 2025-11-21
>
> >**Questions:**
>
> >**Q1: How does the proposed multi-synaptic connectivity fundamentally differ from a per-parameter importance score or learning a supermask over a fixed, overparameterized network？What specific capabilities does the explicit modeling of multiple, independent synapses enable that these other approaches do not?**
>
> >**Q1.1: The differences of multi-synaptic connectivity**
>
> Multi-synaptic connectivity and techniques such as assigning per-parameter importance scores or learning a supermask over a fixed, overparameterized network **operate at different levels and are therefore compatible**.
> Our study specifically focuses on the multi-synaptic cooperation mechanism itself. As described in Section 3.3 (Generalizing to Classic Architecture-Based Methods), when we combine MSCN with architecture-based baselines, we use exactly the same supermask learning rules as baselines. This ensures a fair comparison, so that **the difference only lies in the proposed multi-synaptic cooperation mechanism rather than in the mask learning rules.**
>
> >**Q1.2: The specific capabilities of multi-synaptic structures**
>
> Compared with the baselines, our multi-synaptic cooperation mechanism offers two main advantages.
> - First, the presence of multiple synapses per connection provides stronger plastic capacity, because **each synapse can be optimized independently, which increases the effective capacity of the model without enlarging the backbone**.
> - Second, eligibility traces track local synaptic activity and thereby  modulate plasticity accordingly, enabling task-relevant strengthening or suppression of synapses. This combined design mitigates interference across tasks and improves robustness to task order.
>
> >**Q2:  The precise advantage of this bio-inspired, activity-dependent modulation over a standard gradient-based update. The theoretical or empirical reason why this specific form of modulation is particularly effective for task-order robustness**
>
> Our activity-based modulation is designed to complement the standard gradient-based update by rescaling each synapse’s gradient according to its local activity while preserving the update direction. This **achieves a balance between stability and plasticity at the synaptic level.**
> - When a task is noisy and induces excessively strong synaptic activity, the modulation mechanism suppresses the magnitude of weight updates, thereby protecting consolidated knowledge and limiting excessive performance drift.
> - When a synapse exhibits moderate activity, the modulation allows larger updates, which strengthens learning for the current task. In this way, **the modulation reduces the impact of highly noisy tasks on overall performance**, mitigates interference between tasks, and improves robustness to task order. In the revised manuscript, we have added a more detailed explanation of the modulation mechanism to address these concerns.
>
> >**Q3: Was there any exploration into making P adaptive or variable across layers/connections?**
>
> Thanks for your constructive and fruitful comment. In this work we focus on analyzing how the proposed multi-synaptic cooperation mechanism affects computational efficiency and robustness, so **our study of synapse count is intentionally preliminary at its exploration stage and uses a fixed synapse count to isolate its effect**.
> Due to time constraints, we were unable to explore this direction in depth. We are excited to further investigate making the synapse count adaptive or variable across layers and connections in future work. In the Conclusion section of the revised manuscript, we have added a discussion of this direction.

---

> ### Author Response · Authors · 2025-11-21
>
> >**Q4: Comparing against a strong, non-biologically-inspired baseline that also uses a fixed but over-parameterized network**
>
> We have in fact considered and tested this comparison. As shown in Table 3, **WSN serves as a strong non-biologically inspired baseline** that uses a fixed overparameterized backbone and learns a supermask, so it can be viewed as the type of network you referred to.
> To further assess overall behavior across datasets, we analyzed the average performance over the four benchmarks reported in Table 3. As shown in the table below, MSCN achieves notably better average  computational efficiency (the lowest average Training Time) and order robustness (the lowest average AOPD) while also achieving higher accuracy under the same parameter budget. This **indicates that the gains come from the multi-synaptic cooperation, rather than merely from increased capacity.**
>
> Average method performance across PMNIST, 10-split CIFAR-100, TinyImageNet, and 5-Datasets
> | Method  | Average Training Time ↓ | Average ACC ↑ | Average AOPD ↓ |
> |:-:|:-:|:-:|:-:|
> | PackNet | 1.58 ± 0.11| 79.32 ± 0.19 | 4.62 |
> | SupSup  | 1.41 ± 0.09 | 81.22 ± 0.20 | 3.60 |
> | WSN | 1.28 ± 0.06 | 84.62 ± 0.25  | 2.31|
> | **MSCN**|  **1.14±0.06**| **85.17±0.16** | **2.16** |
>
>
> **References:**
>
> [1] Fan L, Shen H, Lian X, et al. A multisynaptic spiking neuron for simultaneously encoding spatiotemporal dynamics[J]. Nature Communications, 2025, 16(1): 7155.
>
> [2] Hofmann M, Becker M F P, Tetzlaff C, et al. Concept transfer of synaptic diversity from biological to artificial neural networks[J]. Nature communications, 2025, 16(1): 5112.
>
> [3] Zhou C, Liu Y, An X, et al. Optimization of deep learning architecture based on multi-path convolutional neural network algorithm[J]. Scientific Reports, 2025, 15(1): 19532.

---

> > ### Comment · Reviewer_CnT3 · 2025-11-27
> >
> > Thank you for your rebuttal. Having reviewed it, I feel that my original rating still accurately reflects my assessment, and I will not be making any changes.

---

> ### Author Response · Authors · 2025-11-27
>
> Dear Reviewer CnT3,
>
> Thank you for your response and for the time and effort you have devoted to reviewing our manuscript. We appreciate the opportunity your feedback has given us to clarify and improve our work. We respect your decision to maintain your original positive score and sincerely thank you for your support, and wish you a nice and pleasant day :-)

---

### Official Review · Reviewer_jVm8 · 2025-11-01

**Soundness:** 3
**Presentation:** 2
**Contribution:** 3
**Rating:** 4
**Confidence:** 3

**Summary:**

This paper proposes a continuous learning framework for multi-synaptic cooperative networks (MSCN), inspired by the structural and plasticity mechanisms of multi-synaptic connections in biological neural systems. By introducing multi-synaptic connections within a fixed network architecture and integrating plasticity regulation based on local synaptic activity, MSCN aims to enhance the model's representational capacity, resistance to forgetting, and robustness to task sequence variations in continuous learning tasks. Experiments conducted across multiple benchmark datasets—including PMNIST, CIFAR-100, TinyImageNet, and 5-Datasets—covering both spiking neural networks (SNNs) and artificial neural networks (ANNs) demonstrate that MSCN outperforms existing state-of-the-art methods in terms of accuracy and forgetting suppression

**Strengths:**

1. This paper introduces and models the multi-synaptic coordination mechanism in continuous learning for the first time. Unlike traditional network expansion or pruning methods, MSCN enhances the overall model's capacity by strengthening the representational power of individual connections.

2. The paper has a clear structure, with the Methods section describing the implementation of both SNN and ANN.

3. This work provides new insights for bio-inspired artificial intelligence models.

**Weaknesses:**

1. Table 1 compares MSCN with various baseline methods such as EWC, GPM, PackNet, SupSup, etc. However, the paper does not explicitly state whether all baseline methods were run under the identical “multi-head, task label known” setting.Regularization-based and replay-based methods are typically employed more frequently in “class-incremental learning” settings. Comparing their performance in this context to architecture-based approaches (PackNet, SupSup, WSN, MSCN) that achieve BWT=0 in “task-incremental learning” settings is profoundly unfair and exaggerates the latter's advantages.

2. The core modulation function f_mod (Equation 7) in this paper is directly referenced from the work of Zhang et al. (2023). While its integration with a multi-synaptic architecture constitutes a contribution, the paper fails to clearly delineate in its presentation which elements are cited and which represent original innovations.

**Questions:**

1. Can the author clearly specify the specific experimental settings for all comparison methods?

2. The paper demonstrates MSCN's robustness to task orderings. How does the overlap degree of synaptic masks m_j vary under different task sequences? Furthermore, how does the qualification trace f_mod mechanism dynamically preserve or allocate synapses across different task orders?

3. All current experiments are conducted under task-specific incremental learning settings. Could the authors conduct experiments in more realistic and challenging incremental learning scenarios?

---

> ### Author Response · Authors · 2025-11-21
>
> We sincerely appreciate your recognition of our novelty on "multi-synaptic cooperation mechanism in continual learning for the first time", and thank you for your supportive and insightful comments for us to further improve our study.
>
> >**W1.1 & Q1: Experimental setting (whether all methods use the identical setting)**
>
> Yes, all methods (including baseline ones and ours) in Table 1 were run under the same identical setting for fairness. Specifically, all comparison methods and ours were examined under the identical multi-head setting where task identities are assumed to be known at test time. We have revised  the manuscript to explicitly specify this in the experimental setup section of the Appendix.
>
> >**W1.2 Fairness of comparison**
>
> We appreciate the reviewer’s concern regarding the fairness of comparing BWT across different families of continual learning methods. We agree that architecture-based approaches often achieve inherently better BWT than regularization- or replay-based methods, which partly accounts for the observed differences. However, the primary purpose of Table 1 is to provide a comprehensive comparison with recent state-of-the-art methods under **both accuracy and BWT**. Among these metrics, **accuracy**-the central measure of task performance-plays a **more decisive role** in evaluating the effectiveness of the learning system.
>
> As stated in our Related Work section (lines 135–138), we adopt an architecture-based framework to better study its capacity efficiency and robustness, but the proposed multi-synaptic mechanism **is not restricted to this category**. It can be readily integrated into other types of continual learning approaches and is expected to enhance their performance as well. To further support this point, we conducted additional experiments where we combined our multi-synaptic mechanism with methods of different types while keeping the parameter budget unchanged. The results, provided in the table below, **show that incorporating our multi-synaptic mechanism consistently improves accuracy**. Notably, combining our MSCN with ER yields a 1.06% gain on 10-split CIFAR-100, illustrating both the generality and the versatility of our mechanism beyond a specific CL method or experimental setting.
>
> |Type|Method|10-split CIFAR-100|TinyImageNet|5-Datasets|
> |-|-|-|-|-|
> |regularization-based | EWC [1]| 72.77 ± 0.57| 64.51 ± 0.44| 88.64 ± 0.26|
> |regularization-based | **EWC+MSCN**  | **73.26 ± 0.66**   | **64.98 ± 0.54** | **89.23 ± 0.41** |
> |replay-based|ER [2]| 70.07 ± 0.73| 48.32 ± 0.91| 90.11 ± 0.47|
> |replay-based| **ER+MSCN**| **71.13 ± 0.62**| **49.26 ± 0.84** | **90.78 ± 0.38** |
> | architecture-based| Bayesian [3] | 75.57 ± 0.38 | 73.93 ± 0.36 | 93.36 ± 0.31|
> |architecture-based| **Bayesian+MSCN** | **76.48 ± 0.34** | **74.56 ± 0.33** | **94.07 ± 0.27** |
>
> We hope that the above clarification and additional experiments address your concern regarding fairness and further strengthen the evidence for the effectiveness of our method.

---

> > ### Comment · Reviewer_jVm8 · 2025-11-23
> >
> > Thank you very much for answering my questions. I also agree with the contributions of this work. However, I feel that the authors did not directly address my concern regarding the unfair evaluation setup of BWT.
> > In addition, while I understand that accuracy is an appropriate metric for evaluating model performance, I am still very interested in seeing how the model performs in terms of BWT.
> > Even if the results are not ideal, I hope the authors can analyze the specific reasons. If the explanation is reasonable, I would be willing to increase my score; otherwise, I will keep my current rating.

---

> ### Author Response · Authors · 2025-11-21
>
> >**W2: Clear citation and originality**
>
> Thanks for your advice, and accordingly we have added an explicit citation to Zhang et al. (2023) when introducing the modulation function. We appreciate the reviewer’s request to clearly distinguish cited components from our original contributions. We would like to emphasize that our modulation function **is only inspired** by Zhang et al. (2023); **beyond its integration with the multi-synaptic mechanism**, its mathematical form and implementation are **fully redesigned** in our work, with several key differences:
>
> - The modulation function $f_{\text{mod}}$ in Eq. (7) operates on a synapse-specific eligibility trace, and both its concrete **mathematical form and derivation are newly proposed** in our work.
> - Our method emphasizes **local modulation based on synapse activity**, in contrast to Zhang et al. (2023) which performs global modulation driven by the model’s input–output behavior.
> - Our modulation is **targeted at suppressing noise induced by different task orders**, thereby mitigating inter-task interference and improving task-order robustness; in contrast, the modulation in Zhang et al. (2023) is primarily designed to alleviate forgetting.
>
> To avoid any potential confusion, we have further clarified in the revised manuscript how our method differs from that of Zhang et al. (2023),and thus to better represent our original innovations as you suggested.
>
> >**Q2.1: How does the overlap degree of synaptic masks $m_j$ vary under different task sequences?**
>
> For this insightful comment, our short answer is that **masks for tasks that are close in the sequence typically exhibit higher overlap**.
>
> We conducted additional experiments to analyze and show how mask overlap changes across different task sequences. Due to space limitations, here we report only a representative subset of task orders (e.g., `Task#`), and the results of the overlap degree are summarized in the table below.
>
> || Task2 | Task4 | Task6 | Task8 |
> |-|:-:|:-:|:-:|:-:|
> | Task2 | 1.000 | 0.651 | 0.545 | 0.482 |
> | Task4 | 0.651 | 1.000 | 0.698 | 0.639 |
> | Task6 | 0.545 | 0.698 | 1.000 | 0.722 |
> | Task8 | 0.482 | 0.639 | 0.722 | 1.000 |
>
> Here values closer to 1 indicate greater mask overlap between two tasks (row versus column), while smaller values indicate less overlap. We observe that mask **overlap is higher for neighboring tasks**. This is because, in sequential training, masks are selected according to learned relevance scores, and neighboring tasks produce very similar scores over weights, which leads to greater overlap between their masks. As the sequential distance between tasks increases, the relevance scores diverge and the overlap decreases. This observation remains consistent across different task orders.
>
> >**Q2.2: How does the $f_{\text{mod}}$ mechanism preserve or allocate synapses across different task orders?**
>
> This question appears to stem from a possible misunderstanding that we would be glad to clarify here: whether synapses are **preserved or allocated is determined by the mask learning rule described above, not by $f_{\text{mod}}$.**
> The eligibility-trace-based modulation $f_{\text{mod}}$ adjusts synaptic plasticity according to local activity, strengthening or suppressing updates when appropriate, but it does not govern synapse preservation or allocation. We have clarified this distinction in the revised manuscript to avoid any potential confusion.
>
> >**Q3: Conduct experiments in more realistic and challenging incremental learning scenarios**
>
> We sincerely appreciate this thoughtful suggestion. Our study **focuses on analyzing how multi-synaptic cooperation affects capacity efficiency and robustness**. To isolate this mechanism and ensure a fair comparison, **we adopt the same task-incremental learning settings as our baselines** and evaluate all methods under identical assumptions.
> We agree that more realistic and challenging continual-learning scenarios, such as settings without task labels and with blurred task boundaries, are valuable and complementary. Given the time constraints, we have clarified this scope in the paper and discussed how our MSCN can be instantiated under these protocols. We will pursue a comprehensive evaluation in these scenarios in subsequent work to further clarify the performance of the proposed mechanism in more complex settings. Thanks for your valuable comments.
>
> **References:**
>
> [1] Kirkpatrick J, Pascanu R, Rabinowitz N, et al. Overcoming catastrophic forgetting in neural networks[J]. Proceedings of the national academy of sciences, 2017, 114(13): 3521-3526.
>
> [2] Chaudhry A, Rohrbach M, Elhoseiny M, et al. On tiny episodic memories in continual learning[J]. arXiv preprint arXiv:1902.10486, 2019.
>
> [3] Thapa J, Li R. Bayesian adaptation of network depth and width for continual learning[C]//International Conference on Machine Learning. PMLR, 2024, 235: 48038-48061.

---

> ### Author Response · Authors · 2025-11-24
>
> Thanks for your recognition of our contributions.
>
> We are very pleased to report the new and exciting results obtained following your constructive follow-up comments regarding BWT. We fully agree that our previous response did not sufficiently address this aspect beyond the architecture-based comparison, as we initially focused on accuracy—as is common in prior works [1,2].
>
> In response to your explicit request to examine BWT, we conducted additional experiments on both 10-split CIFAR-100 and TinyImageNet under the same parameter budget for all three types of approaches. We are grateful for your suggestion, as these new evaluations further highlight the significance of our contributions **beyond the previously focused architecture-based** approaches.
>
> The results demonstrate not only the previously reported accuracy improvements but also **substantial gains in BWT for both regularization and replay-based methods**, confirming that our MSCN effectively reduces interference and stabilizes prior knowledge.
>
> The experimental results are presented in the table below.
>
> | Type                 | Method            | 10-split CIFAR-100 ACC ↑ | 10-split CIFAR-100 BWT ↑ | TinyImageNet ACC ↑ | TinyImageNet BWT ↑ |
> | -------------------- | ----------------- | ------------------------ | ------------------------ | ------------------ | ------------------ |
> | regularization-based | EWC               | 72.77 ± 0.57             | -3.59±0.49               | 64.51 ± 0.44       | -0.04±0.03         |
> | regularization-based | **EWC+MSCN**      | **73.26 ± 0.66**         | **-2.78±0.19**           | **64.98 ± 0.54**   | **-0.03±0.01**     |
> | replay-based         | ER                | 70.07 ± 0.73             | -7.70±0.59               | 48.32 ± 0.91       | -19.86±0.70        |
> | replay-based         | **ER+MSCN**       | **71.13 ± 0.62**         | **-5.24±0.51**           | **49.26 ± 0.84**   | **-9.17±0.55**     |
> | architecture-based   | Bayesian          | 75.57 ± 0.38             | 0.00±0.00                | 73.93 ± 0.36       | 0.00±0.00          |
> | architecture-based   | **Bayesian+MSCN** | **76.48 ± 0.34**         | **0.00±0.00**            | **74.56 ± 0.33**   | **0.00±0.00**      |
>
> The results show that, **when incorporating MSCN into regularization-based and replay-based methods, it almost always improves BWT**. Notably, combining our MSCN with ER on TinyImageNet improves BWT by 53.8% (an absolute decrease of 10.69).
>
> The observed BWT improvements are due to the high capacity efficiency of our MSCN, which **arises from the independent optimization of multiple parallel synapses**, as demonstrated in Fig. 7. Such higher capacity efficiency has been shown to reduce catastrophic forgetting [3,4,5]. Meanwhile, the modulation mechanism further enhances this property by **depressing the effect of noisy samples and strengthening learning on clean ones**. In contrast, for architecture-based methods, BWT remains zero because the weights of past tasks are frozen, and this result is exactly as expected. At the same time, when our MSCN is incorporated, all three types of methods achieve improved accuracy.
>
> In summary, these additional experiments, conducted in direct response to your thoughtful suggestion, further strengthen our results and highlight the robustness of our MSCN approach.
>
> **References:**
>
> [1] Thapa J, Li R. Bayesian adaptation of network depth and width for continual learning[C]//International Conference on Machine Learning. PMLR, 2024, 235: 48038-48061.
>
> [2] Xiao M, Meng Q, Zhang Z, et al. Hebbian learning based orthogonal projection for continual learning of spiking neural networks[J]. arXiv preprint arXiv:2402.11984, 2024.
>
> [3] Hung C Y, Tu C H, Wu C E, et al. Compacting, picking and growing for unforgetting continual learning[J]. Advances in neural information processing systems, 2019, 32.
>
> [4] Mirzadeh S I, Chaudhry A, Yin D, et al. Wide neural networks forget less catastrophically[C]//International conference on machine learning. PMLR, 2022: 15699-15717.
>
> [5] Farajtabar M, Azizan N, Mott A, et al. Orthogonal gradient descent for continual learning[C]//International conference on artificial intelligence and statistics. PMLR, 2020: 3762-3773.

---

### Author Response · Authors · 2025-12-02
**Summary Comment**

We sincerely thank the new AC for handling our submission, and the reviewers for their feedback and recognition of our work. During the initial review stage, **all reviewers acknowledged the novelty** of our work, which we summarize as follows:

- Inspired by the richness and plasticity of synaptic connections in biological nervous systems, we propose MSCN, the **first** approach that enhances scalability and robustness in continual learning through an explicitly designed **multi-synaptic cooperation** mechanism. (Reviewer jVm8, CnT3, dFvv, M3Db)
- We provide in-depth analyses and experimental evidence that MSCN is **applicable to different types** of continual learning methods and **effectively improves** accuracy, robustness, and BWT. (Reviewer dFvv, M3Db)
- Through extensive experiments, we show that MSCN is applicable to both spiking neural networks (SNNs) and artificial neural networks (ANNs), exhibiting **strong robustness and generalization capability**. (Reviewer dFvv, M3Db)

During the rebuttal phase, we conducted additional experiments and analyses addressing all concerns raised by reviewers. Notably, **some aspects that we initially overlooked were, upon further analysis, found to be our strengths.**

We would also like to express our sincere gratitude to Reviewer **jVm8** for the active engagement in the follow-up discussion, particularly regarding the importance of BWT evaluation. We appreciate the reviewer’s openness to **raising the score** based on further analysis. Motivated by this valuable feedback, we conducted additional experiments that extend beyond the architecture-based comparisons initially emphasized in the submission. The new results turn out to **further reinforce our strength**. The perceived major concerns and our responses are summarized as follows:

- Comparison of BWT: In response to Reviewer jVm8 (W1) and Reviewer M3Db (Q1), we integrated MSCN with different types of continual learning methods (such as regularization-based and replay-based approaches), which effectively improves accuracy, robustness, and BWT, further confirming its generality, scalability, and effectiveness. We are pleased that these results have been recognized by both reviewers and that **Reviewer jVm8 has** **expressed willingness** **to raise** **the rating**.
- Distinction from existing methods: In response to Reviewer jVm8 (W2) and Reviewer CnT3 (W1 & W3), we clarify the biological inspiration of our work and its differences from existing methods, and *emphasize* the multi-synaptic cooperation mechanism as a general strategy applicable to different types of continual learning methods.
- Experimental settings: In response to Reviewer dFvv (Q3) and Reviewer M3Db (Q4), we analyzed the parameter cost of each method, further confirming the performance advantage of our approach under the same parameter budget.

Based on reviewers’ feedback, we have updated our submission (PDF) with the differences highlighted in BLUE. Below is a summary of the key updates:

- Clarification of the experimental setting “multi-head with known task labels”. (Appendix A.2.3)
- Additional experiments integrating our multi-synaptic cooperation mechanism into different types of methods. (Appendix A.3.1)
- A detailed table of notation. (Appendix A.2.5)

We have included all related analyses and discussions in our revised manuscript, and we are confident it is now in its final form. We kindly hope that the new AC will consider the work fairly in light of its novelty, significance, completeness, and strengthened experimental support.

---

### Meta-Review · Area_Chair_qmua · 2026-01-06

**Summary:**

MSCN (Multi-Synaptic Cooperative Networks) is a bio-inspired continual-learning framework that aims to improve capacity and task-order robustness without increasing network depth or width by replacing each connection with multiple parallel synapses and modulating plasticity using eligibility traces derived from recent local activity. The method computes decaying eligibility traces, normalizes them, and applies a nonlinear modulation function to scale gradient updates so that task-relevant synapses are preferentially strengthened while irrelevant ones are suppressed, reducing interference across tasks. MSCN is integrated into a multi-head task-incremental setting via task-specific subnetworks/masks that allocate a portion of parameters to each task and freeze previously used weights. Experiments on PMNIST, 10-split CIFAR-100, TinyImageNet, and 5-Datasets (in both SNN and ANN variants) report higher average accuracy and zero backward transfer compared to several baselines, supported by task-order robustness studies and ablations showing both the multi-synapse structure and modulation are important.

**Reviewer Concerns:**

Overall, reviewers found the idea promising but raised several recurring concerns. First, they questioned the evaluation protocol and fairness of comparisons, especially the emphasis on BWT, given the paper’s multi-head, task-ID-known setting, and asked for results in more realistic continual-learning regimes (e.g., without task labels or under class-incremental/blurred-task conditions). They also felt the novelty and positioning needed tightening: the “first” framing seemed overstated, and the paper should more clearly separate what is borrowed (e.g., the modulation function) from what is genuinely new, and how it differs from related multi-pathway/supermask-style methods. A major theme was scalability and overhead: reviewers wanted clearer accounting of compute/memory/time costs induced by multi-synaptic connectivity and stronger evidence that the approach scales to larger datasets and modern architectures, rather than gains that could be attributed to increased parameterization. Empirically, they noted that improvements can be uneven or modest in some benchmarks and requested a clearer discussion of trade-offs (accuracy vs robustness vs efficiency) and when the method helps most. Finally, they asked for deeper mechanism analysis (why eligibility-modulated updates drive order robustness, how masks behave across task permutations) and for presentation fixes around notation clarity and figure readability.

In the rebuttal, the authors focused on strengthening the empirical and positioning weaknesses raised by reviewers. They added new experiments and analyses in a revised version, most importantly extending MSCN beyond an architecture-only comparison by combining it with both regularization-based and replay-based continual-learning methods and reporting additional results to argue broader applicability and to clarify BWT-related concerns. They also addressed fairness and scalability questions by providing explicit parameter-count tables under a “same parameter budget” setting and adding efficiency and order-robustness summaries (including training-time and order-permutation metrics), aiming to show that improvements stem from the multi-synaptic cooperation mechanism rather than simply increased capacity. Finally, they refined the novelty framing by acknowledging prior multi-synaptic work and clarifying that the central contribution is the continual-learning cooperation mechanism that couples multi-synaptic connectivity with local activity–based plasticity modulation.

**Reviewer Scores:**

Two reviewers rated the paper as marginally above the acceptance threshold, while two rated it as marginally below.

I agree with the reviewers overall and also find the submission borderline. Similar to their feedback, I have concerns about the paper’s positioning, and I encourage the authors to strengthen the camera-ready by clarifying how this work relates to prior neuro-modulation–based approaches, such as Supermasks in Superposition (Wortsman et al., NeurIPS 2020). That said, after reading the reviews and the authors’ rebuttal, I lean toward acceptance.

---

### Decision · Program_Chairs · 2026-01-26

Accept (Poster)